# Conformal Prediction Sets Can Cause Disparate Impact

**Jesse C. Cresswell**
Layer 6 AI
jesse@layer6.ai

**Bhargava Kumar** *
TD Securities
bhargava.kumar@tdsecurities.com

**Yi Sui** *
Layer 6 AI
amy@layer6.ai

**Mouloud Belbahri**
Layer 6 AI
mouloud@layer6.ai

## Abstract

Conformal prediction is a statistically rigorous method for quantifying uncertainty in models by having them output sets of predictions, with larger sets indicating more uncertainty. However, prediction sets are not inherently actionable; many applications require a single output to act on, not several. To overcome this limitation, prediction sets can be provided to a human who then makes an informed decision. In any such system it is crucial to ensure the fairness of outcomes across protected groups, and researchers have proposed that Equalized Coverage be used as the standard for fairness. By conducting experiments with human participants, we demonstrate that providing prediction sets can lead to disparate impact in decisions. Disquietingly, we find that providing sets that satisfy Equalized Coverage actually increases disparate impact compared to marginal coverage. Instead of equalizing coverage, we propose to equalize set sizes across groups which empirically leads to lower disparate impact.

## 1 Introduction

Conformal prediction (CP) (Vovk et al., 2005) has emerged as one of the most promising methods for uncertainty quantification in machine learning because of its wide applicability, and statistical guarantees based on very few assumptions. The main use of CP is to transform heuristic notions of uncertainty into rigorous ones through a calibration step. The output of a conformalized model is a prediction set – a set of likely outputs. To communicate uncertainty, prediction sets are larger when the model is more uncertain about the correct answer.

However, there is a clear drawback to a model that outputs prediction sets. Models are commonly part of decision pipelines where input data is converted into actions. For classification, the possible actions are often mutually exclusive, such that for a given observation we require a single action in response, not a set.

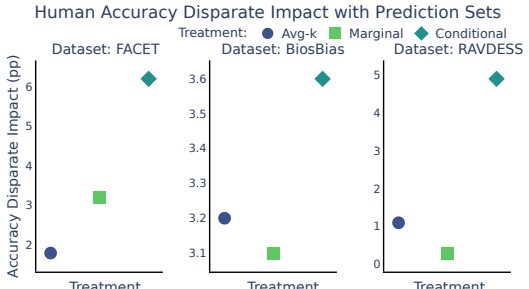

Figure 1: We measure the increase in accuracy per-group from using prediction sets compared to the control population without model assistance. Disparate impact is the maximum difference of increases between groups, and should be minimized for fairness (Equation 7). Prediction sets do not benefit all groups equally, while sets with Equalized Coverage (Conditional) lead to the most unfair outcomes. Statistical analyses and significance are presented in Section 6.

Still, there are many cases where decisions are not left to a model alone, but require a human in the loop. For example, society does not yet use machine learning models to make most medical diagnoses – we prefer human doctors for reasons of safety, accountability, and trust. Nevertheless, there are ample opportunities for people like doctors to use machine learning tools to improve their decisions. As uncertainty quantification is a crucial component for trust, it has been argued that CP sets are a natural fit for such applications (Lu et al., 2022). Indeed, research has shown that providing conformal sets to humans increases their accuracy on decision tasks (Straitouri & Gomez Rodriguez, 2024; Zhang et al., 2024; Cresswell et al., 2024; De Toni et al., 2024).

---

*Equal Contribution

Some of the same research pointed towards a troublesome trend. Cresswell et al. (2024) observed that when conformal sets were given to humans, their accuracy generally increased compared to a control population, but not always by the same amount across groups in the dataset. For some groups accuracy even decreased. This is a major fairness concern. Were a doctor to use prediction sets to assist with diagnoses, it would be unacceptable for some protected groups to have worse outcomes with prediction sets than without.

The fairness of CP has been considered previously by Romano et al. (2020a) who focused on the coverage guarantee that error rates will be no higher than a user-specified tolerance. However, CP controls the error rate only on average across the entire data distribution, and may fluctuate between groups within the distribution. Hence, Romano et al. (2020a) proposed a new fairness standard – that all groups should have Equalized Coverage. Since then, researchers broadly have accepted that equalizing coverage across groups is fair (Lu et al., 2022; Berk et al., 2023b; Ding et al., 2024).

In this work, we focus on the fairness of prediction sets used for human-in-the-loop decision pipelines. Our aim is to provide the first scientific evidence as to the fairness of using CP sets in practical settings. Through pre-registered, randomized controlled trials with human participants, we find that prediction sets can lead to disparate impact – the increases in accuracy compared to the control population are not equal across groups (Figure 1). This alone would be cause for concern, but, distressingly, we further find evidence that applying Equalized Coverage leads to even more unfair outcomes than marginal conformal sets. Hence, we expose a major discrepancy between how fairness has been treated in the literature on CP, and fairness outcomes in practice. We advocate that practitioners move away from Equalized Coverage and instead aim for Equalized Set Size, which correlates strongly with reduced disparate impact.

## 2 BACKGROUND

We begin with background on CP and other set-predictors, and review relevant fairness notions. For the sake of brevity, we only consider the setting of classification in our review and experiments.

### 2.1 CONFORMAL SET PREDICTORS

CP (Vovk et al., 2005; Shafer & Vovk, 2008) is one of the leading approaches to uncertainty quantification in machine learning (Soize, 2017; Abdar et al., 2021), and is aligned with efforts to make models more trustworthy. Other uncertainty quantification approaches often rely on strong assumptions which are unlikely to hold in practice (Gal & Ghahramani, 2016; Lakshminarayanan et al., 2017), or require modifications to the model architecture (Neal, 2012). In contrast, CP applies to black-box models, has no dependence on training data, is distribution free, is valid in finite samples, and assumes only that test data is drawn from the same distribution as calibration data (Vovk et al., 1999; Angelopoulos & Bates, 2021).

Let us consider inputs $x \in \mathcal{X} \subset \mathbb{R}^D$ associated with ground truth classes $y \in \mathcal{Y} = [m]$ where $[m] = \{1, \ldots, m\}$, drawn jointly from a distribution $(x, y) \sim \mathbb{P}$. Given an arbitrary classifier $f : \mathcal{X} \rightarrow [0,1]^m$ with softmax outputs, CP creates a set-predictor (Grycko, 1993), denoted as $\mathcal{C} : \mathcal{X} \rightarrow 2^{[m]}$, with the hallmark property that the sets produced by $\mathcal{C}$ satisfy a coverage guarantee

$$\mathbb{P}[y \in \mathcal{C}(x)] \geq 1 - \alpha, \tag{1}$$

for an error rate $\alpha \in [0, 1]$ which can be chosen *a priori* by the user (Vovk et al., 1999).

CP achieves coverage by allowing the set size $|\mathcal{C}(x)|$ to vary depending on the model's heuristic confidence, e.g., the softmax outputs of $f$. Larger sets indicate greater model uncertainty about $x$. To move from $f$ to $\mathcal{C}$, CP carries out a calibration step on a held-out dataset $\mathcal{D}_{\text{cal}}$ of $n_{\text{cal}}$ datapoints drawn independently from $\mathbb{P}$. First, one defines a *conformal score function* $s : \mathcal{X} \times \mathcal{Y} \rightarrow \mathbb{R}$ with larger scores indicating worse agreement between $x$ and $y$ according to the heuristic uncertainty notion, and then computes $s$ on all datapoints $(x, y) \in \mathcal{D}_{\text{cal}}$. After sorting the scores, one computes the $\frac{\lceil (n+1)(1-\alpha) \rceil}{n}$ quantile which we will denote $\hat{q}$. With the calibration step done, CP sets can be generated for a test datapoint $x_{\text{test}} \sim \mathbb{P}$ as

$$\mathcal{C}_{\hat{q}}(x_{\text{test}}) := \{y \in \mathcal{Y} \mid s(x_{\text{test}}, y) < \hat{q}\}. \tag{2}$$

For any classifier $f$ and any score function $s$, the CP sets $\mathcal{C}_{\hat{q}}$ will provide $1 - \alpha$ coverage (Equation 1).

While valid for any $s$, the usefulness of CP greatly depends on how $s$ is designed as well as properties of the model $f$ including its accuracy and calibration (Guo et al., 2017). The research community has taken great interest in devising score functions that produce sets with the smallest average size. As long as coverage is maintained, small sets are viewed as more useful for applications including human decisioning (Cresswell et al., 2024). Some of the most efficient score functions include APS (Romano et al., 2020b), RAPS (Angelopoulos et al., 2021), and SAPS (Huang et al., 2024).

## 2.2 Conditional Coverage & Mondrian Conformal Prediction

The coverage guarantee in Equation 1 is not totally satisfactory since it holds *marginally* across the entire distribution $\mathbb{P}$. Some subgroups within the distribution may receive lower coverage than $1-\alpha$, for example minority groups which are less well represented in the training set of $f$ leading to lower model accuracy or worse calibration. Ideally, the coverage guarantee would hold for every subgroup of the distribution, down to individual datapoints – so-called conditional coverage (Foygel Barber et al., 2020), but unfortunately this has been proved impossible without distributional assumptions (Vovk, 2012; Lei & Wasserman, 2013). A less strict, and actually attainable goal is group-wise conditional coverage for a pre-specified grouping $g : \mathcal{X} \rightarrow \mathcal{G} = [n_g]$, where each datapoint is assigned to one of $n_g$ groups $a \in \mathcal{G}$, and the coverage guarantee applies to each group separately as

$$\mathbb{P}[y \in \mathcal{C}(x) \mid g(x) = a] \geq 1 - \alpha, \quad \forall\, a \in \mathcal{G}. \tag{3}$$

Group-wise conditional coverage can be achieved through Mondrian CP (Vovk et al., 2003; 2005), which simply partitions $\mathcal{D}_{\text{cal}}$ by groups and performs CP as described in Subsection 2.1 on each group independently. This method can suffer from high variance when $n_g$ is large, or individual groups have scant representation in $\mathcal{D}_{\text{cal}}$, as the effective size of each calibration set could become too small (Angelopoulos & Bates, 2021). Methods to circumvent these issues have been explored (Romano et al., 2020c; Gibbs et al., 2023; Ding et al., 2024).

## 2.3 Non-conformal set predictors

While CP is praised for its distribution-free coverage guarantee, it is not the only way to generate high-quality prediction sets (Chzhen et al., 2021). One other method, average-$k$ set prediction (Algorithm 1), optimizes for the lowest error rate given the constraint $\mathbb{E}|\mathcal{C}| = k$, where $k \in \mathbb{R}^+$. Avg-$k$ prediction relies on a calibration step where a quantile $q_k$ of softmax scores is chosen such that $1-k/m$ of the scores in the calibration set are below $q_k$. Then, prediction sets are formed as

$$\mathcal{C}_{q_k}(x_{\text{test}}) := \{y \in \mathcal{Y} \mid f(x_{\text{test}})_y > q_k\}. \tag{4}$$

---

**Algorithm 1:** Average-$k$ set prediction

**Input:** Calibration dataset $\mathcal{D}_{\text{cal}}$, classifier $f$, average set size $k$, test datapoint $x_{\text{test}}$.

$Y \leftarrow [\,]$
**for** $(x, y) \in \mathcal{D}_{\text{cal}}$ **do**
$\quad Y.\texttt{extend}(f(x))$ // `softmax`
$m \leftarrow \texttt{numClasses}(\mathcal{D}_{\text{cal}})$
$p \leftarrow 1 - k/m$
$q_k \leftarrow \texttt{quantile}(Y, p)$
$y_{\text{test}} \leftarrow f(x_{\text{test}})$ // `softmax`
**return** $y_{\text{test}} > q_k$

---

Optionally, labels $y$ such that $f(x_{\text{test}})_y = q_k$ can be randomly added. The avg-$k$ predictor has optimal error rate under the average size constraint (Lorieul et al., 2021). Since $k$ is allowed to range over positive real values, a target coverage level $1-\alpha$ can be set by computing $q_k$ for different $k$ on a calibration set, and evaluating the empirical coverage on a validation set as

$$1 - \hat{\alpha}_k = \frac{1}{n_{\text{val}}} \sum_{i}^{n_{\text{val}}} \mathbb{1}[y_i \in \mathcal{C}_{q_k}(x_i)]. \tag{5}$$

While the avg-$k$ predictor does not have all the features of conformal methods, it still quantifies the uncertainty of $f$ via variable size prediction sets where larger sets indicate greater uncertainty.

## 2.4 Fairness of set predictors

In real-world applications of machine learning systems, ensuring fairness is of paramount importance, especially in highly regulated industries such as healthcare and financial services. The topic of fairness is incredibly nuanced and cannot be distilled down into a simple set of rules or criteria that should be followed in all cases. Still, it can be useful to delineate high-level approaches.

Two distinct concepts often guide regulatory frameworks: procedural and substantive fairness. Procedural fairness focuses on the integrity of the processes involved in developing models and ensures that all subjects are treated in the same way (Grgić-Hlača et al., 2016). A simple example of procedural fairness in machine learning is when group identifiers (and proxies thereof) are scrubbed from

a dataset. Hence, any model trained on the remaining data cannot rely on group information at all – so called *fairness through unawareness* (Zemel et al., 2013; Kusner et al., 2017).

In contrast, substantive fairness focuses on outcomes, aiming for equitable decisions even if groups are not always treated the same (Dwork et al., 2012). Aligned notions include Equalized Odds and Equalized Opportunity which explicitly use group identifiers to ensure that beneficial outcomes are equally likely across groups (Hardt et al., 2016). Substantive fairness can be measured by computing *disparate impact* for a given metric related to outcomes (Feldman et al., 2015; Esipova et al., 2023). For our set predictors, the relevant metric is the accuracy of humans on a classification task, and hence we consider the change in per-group accuracy when prediction sets are supplied to humans compared to unassisted decisions,

$$\delta_{t,a} := \mathrm{acc}_t[x \in \mathcal{D} \mid g(x) = a] - \mathrm{acc}_{\mathrm{control}}[x \in \mathcal{D} \mid g(x) = a], \tag{6}$$

where $t \in \mathcal{T}$ denotes the *treatment* indicating which set prediction method was used. The disparate impact of providing prediction sets can be written as

$$\Delta_t := \max_{a,b \in \mathcal{G}} (\delta_{t,a} - \delta_{t,b}), \tag{7}$$

where the maximization is done over all pairs of groups. A treatment which has the same beneficial effect on all groups achieves zero disparate impact and is substantively fair. When comparing other quantities between groups, like coverage or set size, we use a similar $\Delta$ notation although the comparison is not with reference to a control treatment, e.g.

$$\Delta_{\mathrm{Cov}} := \max_{a,b \in \mathcal{G}} \left( \mathbb{P}[y \in \mathcal{C}(x) \mid g(x) = a] - \mathbb{P}[y \in \mathcal{C}(x) \mid g(x) = b] \right). \tag{8}$$

The main definition of fairness used in the context of CP is Equalized Coverage (Romano et al., 2020a) which says that all protected groups in the dataset should receive the same level of coverage. Equalized Coverage, expressed formally as $\Delta_{\mathrm{Cov}} \approx 0$, is simply group-wise conditional coverage (Subsection 2.2) over protected groups. Researchers have widely adopted the notion that equalizing coverage is sufficient for ensuring fairness, e.g. Ding et al. (2024) state that with Equalized Coverage "prediction sets ... are effectively 'fair' with respect to all classes, even the less common ones". In our context it is clear that Equalized Coverage aligns with the concept of procedural fairness. Coverage is not an outcome – it is a byproduct of how CP quantifies uncertainty about some underlying task. As we shall demonstrate, equalizing coverage does not translate into fair outcomes on the underlying tasks, in fact it exacerbates disparate impact.

## 3 RELATED WORK

Investigations of the fairness of CP methods began with Equalized Coverage which has remained the standard framework. Romano et al. (2020a) presented a case study where Mondrian CP achieved Equalized Coverage on a medical dataset at the cost of an increased discrepancy between per-group set sizes. Recently, Wang et al. (2024) proposed Equalized Opportunity of Coverage, which takes account of the class label when equalizing coverage between groups. Since its focus is on coverage and not actual outcomes, it still falls under the notion of procedural fairness.

Several works have applied CP to sensitive datasets where fairness considerations are necessary (Lu et al., 2022; Kuchibhotla & Berk, 2023; Berk et al., 2023a). For instance, Lu et al. (2022) discussed how conformal sets could assist medical doctors and specified Equalized Coverage as a desirable prerequisite, although no experiments with doctors were conducted. Straitouri et al. (2024) considered how restricting humans to choose their answer only from a prediction set can cause harm. Other researchers aspired to make CP more fair, such as Liu et al. (2022) who applied demographic parity to conformalized quantile regression (Romano et al., 2019), or Deng et al. (2023) who generalized multicalibration (Hebert-Johnson et al., 2018) to conformal methods, and Zhou & Sesia (2024) who adaptively identified under-covered groups and increased their set sizes to achieve Equalized Coverage.

Researchers have also considered the fairness of uncertainty quantification methods beyond set-predictors. Calibration is an essential prerequisite for uncertainty quantification which stands in tension with fairness requirements (Pleiss et al., 2017). Ali et al. (2021) advocate that fairness approaches should focus on errors from epistemic uncertainty, Kuzucu et al. (2023) identify that models may appear fair according to point predictions while providing biased uncertainty estimates, and Mehta et al. (2024) explore the impact of applying fairness methods on uncertainty estimates.

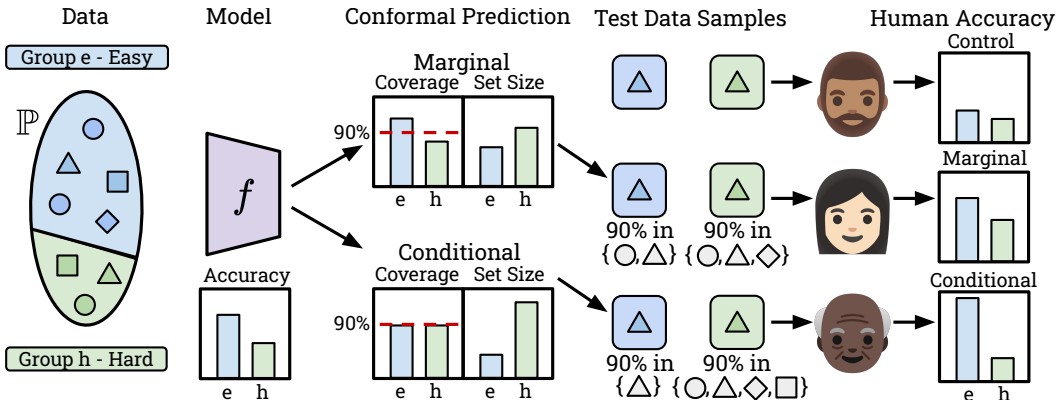

Figure 2: Illustration of how unfairness can arise in CP. Given a data distribution $\mathbb{P}$ with groups of differing difficulty, a model $f$ may have inherent bias. Using marginal CP can translate to lower coverage and larger sets for the harder group. To equalize coverage, conditional CP must increase set sizes on the harder class, and reduce them on the easier class. Since human accuracy correlates strongly with set size, not coverage, outcomes become more unfair with Equalized Coverage.

## 4 HYPOTHESES & METHOD

### 4.1 HYPOTHESES

Suppose $f$ was trained on a dataset with some inherent bias, or that groups naturally vary in difficulty such that $f$ has different accuracy on two groups $e$ and $h$ ("easy" and "hard"). Marginal CP using a sensible score function will tend to overcover examples from $e$ and undercover examples from $h$. To achieve group-wise conditional coverage instead of marginal, the coverage on $e$ must decrease, while the coverage on $h$ increases. For a fixed model $f$ and score function $s$, the only way to increase coverage on $h$ is to increase the number of classes added to prediction sets for $h$ – in other words, average set size must increase for $h$, and correspondingly decrease for $e$. If set sizes were comparable across groups for marginal sets, then conditionally covered sets will tend to have more unequal average set sizes across groups. Essentially, there is a tradeoff between equalizing coverage, and equalizing set size.

On its own this tradeoff is not directly connected to the substantive fairness of using prediction sets. However, it has been observed in previous studies that it is set size, not coverage, which has the greatest influence on human accuracy, and therefore on outcomes (Cresswell et al., 2024). Together, these facts imply that equalizing coverage will create more imbalanced set sizes which will propagate into more biased human performance. The proposed mechanism by which equalizing coverage with Mondrian (conditional) CP causes disparate impact is illustrated in Figure 2.

Based on the above discussion, we propose two hypotheses:

**Hypothesis 1** *Prediction sets supplied to human decision makers can cause disparate impact in the human's performance.*

In particular, if a model has biased performance across groups, marginal CP will tend to produce unequal set sizes which will in turn provide uneven utility to human decision makers.

**Hypothesis 2** *Prediction sets created with Mondrian CP to ensure Equalized Coverage will cause greater disparate impact than marginal prediction sets.*

The use of Mondrian CP or similar methods to equalize coverage will benefit groups that have higher model accuracy more than groups with low accuracy, producing a Matthew effect where "the rich get richer", instead of a more equitable Robin Hood effect (Ganev et al., 2022).

The remainder of this paper investigates whether these hypotheses can be supported by data, along with the claim that set size has greater influence on human accuracy than coverage.

## 4.2 METHOD

We conduct randomized controlled trials with human subjects to discern the impact of various set prediction methods on per-group accuracies, and measure potential disparate impacts. Our hypotheses, experiments, and analysis plans were pre-registered.[1] In each experiment, paid human volunteers are asked to complete a challenging task and are provided prediction sets from a trained machine learning model (or for the control, no assistance is given). The sets are constructed either with marginal coverage guarantees (Equation 1), group-wise conditional coverage through Mondrian CP on a specified grouping (Equation 3), or via the avg-$k$ method (Equation 4). In all cases the same model is used, aiming for the same coverage level over $\mathcal{D}_{\text{cal}}$. We call these four options *treatments*, and refer to them as control, marginal, conditional, and avg-$k$.

We conduct independent experiments on three distinct tasks. For each task, $N$ unique participants are recruited and are randomly but evenly partitioned into the four treatments. During the experiments we conduct $M$ sequential *trials*, each where the participant is shown a datapoint from $\mathcal{D}_{\text{test}}$, drawn i.i.d. from the same distribution as $\mathcal{D}_{\text{cal}}$, along with a prediction set and a stated coverage guarantee (except for the control). No other information from the model such as softmax scores is given. The tasks are each designed as forced choices where participants must select one class $y$ out of all $m$ available classes, and only one class is considered correct.

Given data collected in this way we can compute the accuracy improvements (Equation 6) and disparate impact (Equation 7) for each treatment directly. However, this direct analysis would not account for the design of the experimental method which could result in overconfident inferences (Yarkoni, 2022). In particular, each participant provides multiple responses, participants may have different inherent ability levels on their task, and samples drawn from $\mathcal{D}_{\text{test}}$ can also have different inherent difficulties. Hence, our formal statistical analysis models human performance on the tasks using Generalized Estimating Equations (GEEs).[2] GEEs can be seen as Generalized Linear Models for clustered responses that take into account intra-participant response correlations (Liang & Zeger, 1986). Hence, GEEs are a suitable choice since each participant represents a cluster of responses.

Formally, we denote the response variable by $\text{correct}_{i,j}$, indicating whether participant $i$ selected the correct label for trial $j$. Additionally, we consider the following covariates: (i) $\text{treat}_i$ denoting the treatment participant $i$ is randomly assigned to; (ii) $\text{group}_{i,j}$ which is the protected group for trial $j$ that participant $i$ sees; and (iii) $\text{diff}_{i,j}$, the marginal conformal set size of trial $j$ shown to participant $i$, which acts as a proxy for the inherent difficulty of that datapoint.

For each task, we fit GEEs to estimate a marginal model for the effect of $\text{treat}$ on the participants $\text{correct}$ responses, controlling for $\text{group}$ and $\text{diff}$ covariates. The model can be expressed as

$$\text{logit}(\mathbb{E}[\text{correct}_{i,j}]) \sim \text{treat}_i \times \text{group}_{i,j} + \text{diff}_{i,j}, \tag{9}$$

for $i = 1, \ldots, N$ and $j = 1, \ldots, M$, where $\text{logit}(x) = \log \frac{x}{1-x}$, and the notation $A \times B$ indicates an interaction between covariates $A$ and $B$.

Alongside empirical estimates of the standard errors, it is straightforward to derive odds ratios (ORs) from the GEE model. ORs allow us to estimate how much more likely a participant in treatment $t$ is to give the correct answer than if they were in the control, which is another way of expressing the expected accuracy improvement due to $t$ (c.f. Equation 6). For the sake of clarity, we give the formal description below. Let $p_{t,a} := \hat{\mathbb{P}}(\text{correct} = 1 \mid \text{treat} = t, \text{group} = a)$ be the probability of a correct answer under the GEE model, conditional on treatment and group. For treatment $t$, and for every protected group $a$, the OR of $t$ versus control is given by

$$\mathbf{OR}_{t,a} := \frac{p_{t,a}/(1 - p_{t,a})}{p_{\text{control},a}/(1 - p_{\text{control},a})}. \tag{10}$$

The interpretation of $\mathbf{OR}_{t,a}$ is simple. For example, $\mathbf{OR}_{t,a} = 1.4$ means that for trials representing group $a$, the odds for participants assigned to treatment $t$ to give the correct answer are 40% higher compared with participants assigned to the control treatment.

To assess the disparate impact caused by any treatment for Hypothesis 1, we can compare ORs between groups. We define the ratio of ORs (RORs) for treatment $t$ and groups $a$ and $b$ as

---

[1]The pre-registration is viewable at `osf.io/75hm9`.

[2]We use the GEE implementation from the `statsmodels` python package.

$\mathbf{ROR}_{t,a,b} := \mathbf{OR}_{t,a}/\mathbf{OR}_{t,b}$. When $\mathbf{ROR}_{t,a,b} \approx 1$, treatment $t$ provides the same advantage over the control for group $a$ as it does for group $b$ which is a fair outcome. Hence, a value of $\mathbf{ROR}_{t,a,b}$ greater than 1 for some pair $(a,b)$ indicates that $t$ causes disparate impact, and so we report the maximum ROR between pairs of groups as a descriptive statistic (c.f. Equation 7),

$$\mathbf{maxROR}_t := \max_{a,b \in \mathcal{G}} \mathbf{ROR}_{t,a,b}. \tag{11}$$

Finally, to assess whether equalizing coverage increases unfairness for Hypothesis 2, we compare $\mathbf{maxROR}_{\text{Marg}}$ to $\mathbf{maxROR}_{\text{Cond}}$. If the latter value exceeds the former, it indicates that equalizing coverage amplifies disparate impact relative to the marginal treatment.

## 5 EXPERIMENTS & EVALUATION

### 5.1 TASKS, DATASETS, AND MODELS

Using open-access datasets from the machine learning fairness literature, we created three tasks where human decision makers could potentially take advantage of model assistance.

**Image Classification**  Image classification is performed by humans daily in impactful settings, for example by radiologists who view X-ray images from patients of all ages. For a similar setting we used the FACET dataset (Gustafson et al., 2023) of images of people, labeled by their occupation with grouping by age. We used the 20 most common classes and split the dataset into $\mathcal{D}_{\text{cal}}$, $\mathcal{D}_{\text{calval}}$, and $\mathcal{D}_{\text{test}}$ stratified by class. The age annotations come in four pre-defined groups: Younger, Middle, Older, and Unknown. We used CLIP ViT-L/14 (Radford et al., 2021) as a zero-shot classifier.

**Text Classification**  Text classification is often used to organize large amounts of text data, for instance in recruiting where gender bias can easily manifest. As a surrogate task, we employed the BiosBias dataset (De-Arteaga et al., 2019) which contains personal biographies classified by occupation, and grouped by binary gender. We selected 10 of the most common occupations and then split the dataset into $\mathcal{D}_{\text{train}}$, $\mathcal{D}_{\text{val}}$, $\mathcal{D}_{\text{cal}}$, $\mathcal{D}_{\text{calval}}$, and $\mathcal{D}_{\text{test}}$, ensuring class balance. $\mathcal{D}_{\text{train}}$ and $\mathcal{D}_{\text{val}}$ were used for classifier training, while the remaining splits were used for prediction set construction. For classification, we generated representations of the biographies using a pre-trained BERT model (Devlin et al., 2019; Huggingface, 2024), then trained a linear classifier on these representations.

**Audio Emotion Recognition**  Emotion recognition is performed naturally in human communication, but emotional expression can vary greatly between speakers of different genders. We utilized audio recordings from the RAVDESS dataset (Livingstone & Russo, 2018) where female and male actors convey 8 emotions using the same short phrases. We partitioned the dataset into $\mathcal{D}_{\text{cal}}$, $\mathcal{D}_{\text{calval}}$, and $\mathcal{D}_{\text{test}}$ ensuring stratification by class (emotion) and group (binary gender), and used a fine-tuned wav2vec2 model (Baevski et al., 2020; Fadel, 2023) for emotion classification.

For all three tasks, we aimed to compare disparate impact between avg-$k$, marginal, and conditional prediction sets with target 90% coverage. For greater diversity in approaches to CP, FACET and RAVDESS used the RAPS score function (Angelopoulos et al., 2021), while BiosBias used SAPS (Huang et al., 2024). The hyperparameters of these score functions were tuned on $\mathcal{D}_{\text{calval}}$ with 50 iterations to minimize average set size using Optuna (Akiba et al., 2019a). Then, each method used $\mathcal{D}_{\text{cal}}$ to

Table 1: Model metrics on $\mathcal{D}_{\text{test}}$

| Task | Top-1 | $\Delta_{\text{Top-1}}$ | Method | Cov. | Size | $\Delta_{\text{Cov}}$ | $\Delta_{\text{Size}}$ |
|------|-------|-------|--------|------|------|------|------|
| FACET | 70.0 | 22.2 | Avg-$k$ | 90.2 | 2.52 | 5.8 | 1.10 |
| | | | Marg. | 89.7 | 2.62 | 11.4 | 0.75 |
| | | | Cond. | 90.0 | 2.76 | 2.6 | 3.16 |
| BiosBias | 78.9 | 2.7 | Avg-$k$ | 88.2 | 1.52 | 3.3 | 0.01 |
| | | | Marg. | 89.8 | 1.69 | 2.7 | 0.03 |
| | | | Cond. | 89.1 | 1.70 | 0.0 | 0.43 |
| RAVDESS | 71.1 | 7.8 | Avg-$k$ | 91.4 | 1.97 | 2.8 | 0.12 |
| | | | Marg. | 90.8 | 1.94 | 3.9 | 0.07 |
| | | | Cond. | 91.4 | 2.01 | 0.6 | 0.71 |

calibrate its threshold(s) for set prediction. Table 1 displays metrics computed on $\mathcal{D}_{\text{test}}$ for the models used in each task, including the top-1 accuracy, and for each set prediction method the empirical coverage and average set sizes. We note that all methods achieve coverage close to 90%, while conditional CP tends to have larger set sizes due to its more stringent requirements. Also shown are the maximum per-group differences in model accuracy, coverage, and set size defined as in Equation 8. We note that conditional CP is the only method that achieves Equalized Coverage ($\Delta_{\text{Cov}} \approx 0$), but at the cost of a larger difference in set sizes between groups.

Additional details on tasks, datasets, models, and set prediction methods are given in Appendix A. Our code is available at `github.com/layer6ai-labs/conformal-prediction-fairness`.

## 5.2 EXPERIMENT DESIGN

We created experiments and hosted them online with participants recruited through Prolific (Prolific, 2024). To ensure high-quality data, participants were trained on their specific task and given a financial incentive to answer correctly. Each experiment consisted of a task (FACET, BiosBias, or RAVDESS), and a treatment (control, avg-$k$, marginal, or conditional). We recruited 600 participants ($N = 200$ per task, 50 per experiment), and paid them on average 9.75 GBP per hour, totaling 1500 GBP for participant payment.

During each experiment, participants were first shown a consent form detailing how their data would be collected and used. Then, participants were introduced to the task and trained on 20 practice trials which were not used in our analysis. The testing phase proceeded with $M = 50$ more trials, an ex-

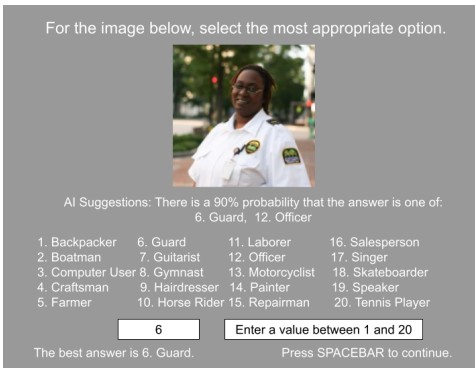

Figure 3: Main trial screen shown to participants for FACET with marginal conformal set treatment. The correct answer is given only after the participant responds.

ample of which is shown in Figure 3 for FACET, while detailed task descriptions can be found in Appendix B. For each trial, participants were presented with a datapoint $x$ along with all $m$ class labels and were asked to classify $x$. For the avg-$k$, marginal, and conditional treatments participants were also shown a prediction set, accompanied by the expected coverage. There was no time limit for responses, however for FACET the image $x$ was only displayed for one second to increase task difficulty. After the class was selected for each trial, the correct answer was revealed.

## 6 RESULTS

### 6.1 DISPARATE IMPACT MEASUREMENT IN HUMAN SUBJECT EXPERIMENTS

Figure 1 directly visualizes the data we collected, showing the disparate impact on accuracy $\Delta_t$ calculated as in Equation 7, treating each trial as an independent observation. We observe that disparate impact is present for most tasks and treatments ($\Delta_t > 0$), notably with the conditional treatment consistently worse. As discussed in Subsection 4.2, each trial is not independent, so we present Figure 1 only for intuition, and not as formal statistical analysis.

Instead, we applied the more rigorous statistical analysis with GEEs and present the results in Table 2 which shows the ORs of human accuracy for each task and treatment as compared to the control, along with maximum RORs across groups. We interpret the findings in the context of our two hypotheses from Subsection 4.1.

Hypothesis 1 supposes that biases from a model can propagate to humans using prediction sets as an aid and result in disparate impact. Table 1 demonstrates that the models we used have some biases across groups, with non-zero $\Delta_{\text{Top-1}}$ values, which translated to non-zero $\Delta_{\text{Cov}}$ for the avg-$k$ and marginal sets. Conditional sets do equalize coverage ($\Delta_{\text{Cov}} \approx 0$), but at the cost of greater discrepancies in set size, $\Delta_{\text{Size}}$. From Table 2 we see that in most cases prediction sets were useful to participants versus the control ($\mathbf{OR}_{t,a} > 1$), supporting previous findings (e.g. Cresswell et al. (2024)). However, it is clear that the beneficial effect is not experienced equally across groups. For FACET, the Younger group had the highest model accuracy (Figure 5), and experienced the most improvement across all treatments. The Older and Unknown groups having lower model accuracy and minority representation (see Table 4 in Appendix A) saw less improvement, and even some harm with conditional sets where human performance decreased ($\mathbf{OR}_{t,a} < 1$). Hence, all three treatments caused disparate impact ($\mathbf{maxROR}_t > 1$). Similar results are seen for BiosBias. Even though the model itself was much less biased (Table 1) and both Female and Male groups benefited when sets were provided, the amount of benefit was not equally shared. Both FACET and BiosBias show that providing prediction sets can cause disparate impact, in support of Hypothesis 1. RAVDESS provides an example that disparate impact need not always occur. Again, both groups benefited from each treatment, but for the avg-$k$ and marginal treatments the increase actually was commensurate across groups. For these two treatments the prediction sets did not have equal coverage, but did have close to the same average set sizes across groups.

Table 2: Summary statistics from the GEE models: $\mathbf{OR}_t$ (Equation 10) where values greater than 1 indicate treatment $t$ had a positive effect on human accuracy for group $a$, and the $\star$ and $\diamond$ superscripts indicate significance at $5\%$ and $10\%$ levels respectively; $\mathbf{maxROR}_t$ (Equation 11), where values greater than 1 indicate disparate impact – treatment $t$ benefited one group more than another.

| Dataset | Group | $\mathbf{OR}_{\text{Avg}-k}$ | $\mathbf{OR}_{\text{Marg}}$ | $\mathbf{OR}_{\text{Cond}}$ | $\mathbf{maxROR}_{\text{Avg}-k}$ | $\mathbf{maxROR}_{\text{Marg}}$ | $\mathbf{maxROR}_{\text{Cond}}$ |
|---|---|---|---|---|---|---|---|
| FACET | Younger | 1.19 | 1.34$^\diamond$ | 1.37$^\star$ | 1.11 | 1.26 | 1.51 |
| | Middle | 1.12 | 1.20$^\star$ | 1.19$^\diamond$ | | | |
| | Older | 1.17 | 1.08 | 0.91 | | | |
| | Unknown | 1.07 | 1.06 | 0.91 | | | |
| BiosBias | Female | 1.91$^\star$ | 1.46$^\star$ | 1.91$^\star$ | 1.34 | 1.12 | 1.33 |
| | Male | 1.42$^\star$ | 1.63$^\star$ | 1.44$^\star$ | | | |
| RAVDESS | Female | 1.32$^\star$ | 1.34$^\star$ | 1.43$^\star$ | 1.02 | 1.01 | 1.28 |
| | Male | 1.30$^\star$ | 1.36$^\star$ | 1.12 | | | |

Hypothesis 2 proposes that equalizing coverage with the conditional treatment will cause more disparate impact than the marginal treatment. Our data in Table 2 strongly supports this conclusion where we find that the value of $\mathbf{maxROR}_{\text{Cond}}$ is consistently higher than $\mathbf{maxROR}_{\text{Marg}}$. Even for RAVDESS where the marginal treatment did not show disparate impact, the conditional treatment introduced noticeable bias.

## 6.2 INSIGHTS

**Key factors impacting the fairness of human performance** We aim to identify the factors contributing to $\Delta_t$, disparate impact in accuracy improvement, as visualized in Figure 1. $\Delta_t$ measures the difference in accuracy improvement over the control group between the most and least improved groups. To investigate this, we plot in Figure 4 the differences between these groups across four key factors: coverage, adoption, average set size, and singleton frequency. Adoption measures the fraction of answers participants selected from the prediction set, while the singleton frequency refers to how often a singleton set was provided, both of which were identified by Cresswell et al. (2024) as factors contributing to how helpful prediction sets are to humans. We note that the differences for these four quantities can be negative because we select the max and min groups based on accuracy improvement – they are not necessarily the max and min groups respectively for the four quantities.

It is clear that the conditional treatment achieves coverage differences that are closest to zero (Equalized Coverage). However, this does not translate to lower $\Delta_t$. Instead, experiments with relatively large (in magnitude) coverage difference, showed the most fair outcomes (e.g. RAVDESS-Marginal where the best and worst groups saw coverage levels differ by 4 percentage points, but still had the same accuracy). Additionally, there is no apparent correlation between coverage difference and $\Delta_t$ which implies coverage differences do not drive human accuracy outcomes. One might suppose that if the participants relied on sets more for some groups than others, disparate impact could arise. However, like for coverage, we see no strong correlation between the adoption difference and $\Delta_t$. The experiment with the single most fair outcomes also had the largest adoption difference.

By contrast, we see strong correlation between set size differences and $\Delta_t$. The most fair outcomes occur when set sizes are similar across all groups, as for RAVDESS-Marginal, or FACET-Avg-K. The conditional treatment which increases set size differences in its effort to equalize coverage also increases disparate impact. Similar to set size, there is also clear correlation between singleton frequency and $\Delta_t$. Cresswell et al. (2024) observed that human accuracy on tasks was highest when the model expressed certainty through singleton sets. Hence, groups with more singletons could expect higher accuracy, so it is plausible that equalizing how often singletons occur would encourage fair outcomes, which is what we observe. Users should be aware that group-conditional modifications such as equalizing singleton frequency are susceptible to Yule effects (Ruggieri et al., 2024).

In summary, our results show that simply equalizing coverage, as is the aim of the conditional treatment, is not an effective way to promote fair outcomes. Instead, focusing on Equalized Set Size or Equalized Singleton Frequency could be more useful standards of fairness for set predictors.

**The role of group difficulty in disparate impact** Figure 5 illustrates accuracy on the FACET dataset across different age groups for both the model and humans (control), and when broken down by treatment. We observe that both the model and humans are consistent in which groups they

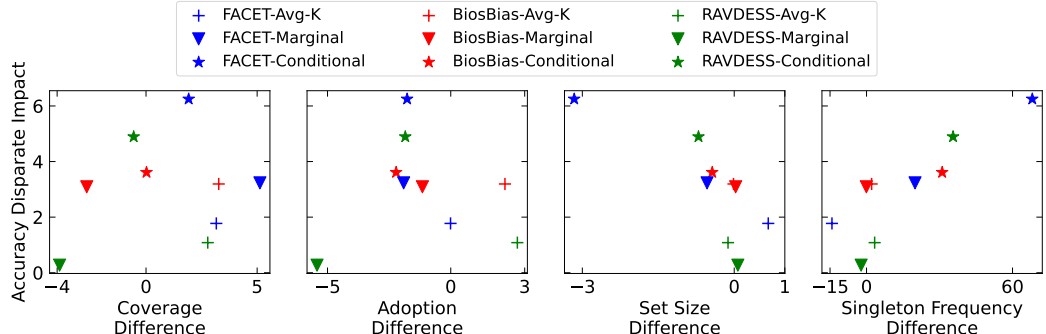

Figure 4: Accuracy disparate impact $\Delta_t$ compared to the difference between the most and least improved groups across key factors for various datasets and treatments. From left to right: coverage, adoption, average set size, and singleton frequency.

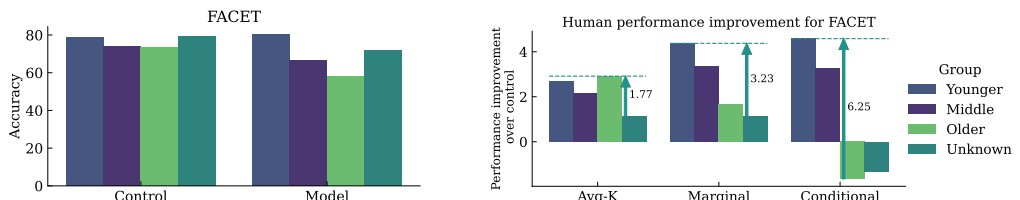

Figure 5: Accuracy by group for FACET. **Left:** Human accuracy (Control) and model Top-1 Accuracy across groups. **Right:** Human accuracy across groups and treatments.

find more or less difficult, with the model showing greater variance between groups. Providing marginal or conditional prediction sets tends to improve human performance more for the group the model had highest accuracy on (Younger), while offering less benefit—and in some cases even worsening performance—for the lowest model accuracy group (Older). This analysis demonstrates how prediction sets may contribute to disparate impact by disproportionately helping easy groups and harming hard groups, widening the gap for outcomes.

## 7 CONCLUSIONS & LIMITATIONS

In this work we have presented the first experimental study on the fairness of conformal prediction sets as a human decisioning aid. Our data runs contrary to the prevailing wisdom in the research community – we find that equalizing coverage (Romano et al., 2020a) across groups actually increases unfairness of outcomes. Based on these results, we instead recommend that practitioners aim for Equalized Set Size or Equalized Singleton Frequency across protected groups, as both set size and singleton frequency correlate much more strongly with outcomes than coverage does.

Still, any study, especially those involving humans, will have its limitations. While we recruited 600 unique participants and collected 30,000 individual responses, some of the statistical measures in Table 2 did not achieve significance at the 5% level due to insufficient observations. This is a consequence of the inherent imbalance between groups in the datasets we used, which is expected to occur in real-world data, and is a crucial aspect for our discussions on fairness. In particular, the FACET dataset was divided into four groups compared to only two for BiosBias and RAVDESS, which means fewer examples from each FACET group were shown to participants, and hence our conclusions have less statistical significance on them. The low significance on individual groups is mitigated by the consistent results we see across three independent datasets, which when taken together provide clear evidence for our hypotheses.

Additionally, our experiments only used classification tasks, whereas conformal prediction has also been applied to regression (Romano et al., 2019), time-series (Stankeviciute et al., 2021), and natural language tasks (Mohri & Hashimoto, 2024). Other instantiations of conformal prediction may be susceptible to the same unfairness mechanism we proposed in Figure 2, where difficult groups tend to be undercovered, requiring larger sets for Equalized Coverage which is a disadvantage, but this is out-of-scope for our study.

**Ethics Statement** Throughout our research, we took steps to ensure that research ethics were upheld. We acknowledge that ethical considerations are extremely important for any research involving human subjects, and we have ensured that our research meets the Code of Ethics for ICLR 2025. We also followed the ethical standards of our institution. Although it does not have an internal review board (IRB) process, the steps we took to meet its standards included: understanding the protocols and procedures for ethical research at our institution; curating datasets that only contained content suitable for showing to participants; performing the experiments on ourselves as authors; providing participants with a clear consent form that detailed the data that would be collected as well as how it would be used and how long it would be retained; paying all recruits a fair wage; and ensuring we were aware of potential biases in the data we collected by reviewing the demographic distribution of recruits (Table 8).

**Reproducibility Statement** We have made two specific efforts to promote the reproducibility of our results. First, we have released our codebase and the data we collected at `github.com/layer6ai-labs/conformal-prediction-fairness`, which will enable other researchers to perform the same set prediction methods, generate the same dataset splits, and perform the same statistical analysis. Second, we have extensively described our experimental setup in Appendix B including images of the screens that were shown to human participants throughout the experiment. The description is complete enough for a third party to faithfully recreate and reproduce our experiment.

ACKNOWLEDGMENTS

We would like to thank Noël Vouitsis for comments on a draft of this manuscript.

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

## A    IMPLEMENTATION DETAILS

Our code for curating datasets and performing calibration and set prediction is available at `github.com/layer6ai-labs/conformal-prediction-fairness`. In this section we detail the steps that were taken.

### A.1    DATASET PRE-PROCESSING

High-level information for the calibration-validation, calibration, and test datasets are shown in Table 3. The following sections provide a detailed explanation of the pre-processing steps applied to each dataset. Dataset pre-processing and set prediction does not require extensive computing resources. We used an Intel Xeon Silver 4114 CPU and TITAN V GPU, which took in total less than 1 hour to process all three datasets.

Table 3: Dataset Information

| Dataset | $|\mathcal{D}_{\text{calval}}|$ | $|\mathcal{D}_{\text{cal}}|$ | $|\mathcal{D}_{\text{test}}|$ | Total Classes | Used Classes ($m$) | Groups ($n_g$) |
|---|---|---|---|---|---|---|
| FACET | 1400 | 4000 | 1400 | 52 | 20 | 4 |
| BiosBias | 5000 | 10000 | 2000 | 28 | 10 | 2 |
| RAVDESS | 240 | 840 | 360 | 8 | 8 | 2 |

Table 4: FACET Group Counts

| Group | $\mathcal{D}_{\text{calval}}$ | $\mathcal{D}_{\text{cal}}$ | $\mathcal{D}_{\text{test}}$ |
|---|---|---|---|
| Younger | 254 | 711 | 276 |
| Middle | 772 | 2144 | 729 |
| Older | 103 | 299 | 91 |
| Unknown | 271 | 846 | 304 |

Table 5: BiosBias Group Counts

| Group | $\mathcal{D}_{\text{calval}}$ | $\mathcal{D}_{\text{cal}}$ | $\mathcal{D}_{\text{test}}$ |
|---|---|---|---|
| Female | 2424 | 4887 | 969 |
| Male | 2576 | 5113 | 1031 |

Table 6: RAVDESS Group Counts

| Group | $\mathcal{D}_{\text{calval}}$ | $\mathcal{D}_{\text{cal}}$ | $\mathcal{D}_{\text{test}}$ |
|---|---|---|---|
| Female | 120 | 420 | 180 |
| Male | 120 | 420 | 180 |

**FACET** The FACET dataset (Gustafson et al., 2023) was created to investigate bias in image classification tasks, with images of people annotated by human labelers, and sensitive attributes including perceived age, gender, and skin tone. We used the occupation annotations as class labels, and age annotations for groups. Age groups are categorized as *'Younger'* (perceived to be $< 25$ years old), *'Middle'* (25 - 65), *'Older'* ($> 65$), and *'Unknown'*. Preprocessing involved filtering for images containing a single person and a single occupation label. Images were resized to $224 \times 224$ pixels and center-cropped for consistency. We focused on the top $m = 20$ most common occupations which were: [*backpacker*, *boatman*, *computer user*, *craftsman*, *farmer*, *guard*, *guitarist*, *gymnast*, *hairdresser*, *horseman*, *laborer*, *lawman*, *motorcyclist*, *painter*, *repairman*, *seller*, *singer*, *skateboarder*, *speaker*, *tennis player*]. Since some of these labels are unintuitive, we clarified them for presentation to participants by renaming them to: [*Backpacker*, *Boatman*, *Computer User*, *Craftsman*, *Farmer*, *Guard*, *Guitarist*, *Gymnast*, *Hairdresser*, *Horse Rider*, *Laborer*, *Officer*, *Motorcyclist*, *Painter*, *Repairman*, *Salesperson*, *Singer*, *Skateboarder*, *Speaker*, *Tennis Player*]. For conformal prediction, the remaining dataset was split into 1400 calibration-validation samples, 4000 calibration samples, and 1400 test samples, using stratified sampling to ensure balanced occupation representation across subsets. Groups, however, were not stratified or balanced, and followed the natural distribution in the dataset (see Table 4). The calibration-validation, calibration, and test sets were treated identically, making them i.i.d. for conformal prediction. FACET is released under a custom license that allows it to be used "for the purposes of measuring or evaluating the robustness and algorithmic fairness of AI and machine-learning vision models, and solely on a non-commercial and research basis".

**BiosBias** The BiosBias dataset (De-Arteaga et al., 2019) contains public biographies labeled by occupation and annotated with binary gender groups. Biosbias created occupation labels automatically by extracting information from the biographies. They also extracted likely binary gender information from 3rd person pronoun use. We used occupations as class labels, with two groups, "Female" and "Male". To preprocess BiosBias data, we removed non-ASCII characters, URLs, emails, phone numbers, and redundant punctuation, then truncated the biographies to 400 characters to limit variance due to vastly different text lengths. We generated text representations of each biography using a pre-trained BERT model (Devlin et al., 2019), and trained a linear classifier to predict occupations using a training set of 50,000 samples with a validation set of 5,000 samples for manual hyperparameter selection and evaluation. From the original 28 occupations, we focused on $m = 10$ of the most common: [*Professor*, *Physician*, *Photographer*, *Journalist*, *Psychologist*, *Teacher*, *Dentist*, *Surgeon*, *Painter*, and *Model*]. These were not strictly the 10 most common occupations since we found that the data quality of some occupations was noticably worse than others; for example we excluded the textitNurse class because it contained hundreds of near-duplicate biographies. For conformal prediction, we used 5,000 calibration-validation samples, 10,000 calibration samples, and 2,000 test samples, using stratified sampling to ensure balanced representation across all occupations. Groups were not stratified, but were already roughly balanced in the dataset which is reflected in our splits (Table 5). This method of splitting ensured the calibration and test sets could be treated as i.i.d. for conformal prediction purposes. BiosBias is released under a MIT license.

**RAVDESS** For RAVDESS (Livingstone & Russo, 2018), a dataset of audio-visual recordings where actors express emotions, we worked with 1,440 audio-only samples exhibiting $m = 8$ emotions: [*Happy*, *Angry*, *Calm*, *Fearful*, *Neutral*, *Disgust*, *Sad* and *Surprised*]. The dataset contains recordings from 24 actors, 12 female and 12 male, so we used emotion as the class label and gender as the protected group. Each actor recorded 60 snippets of approximately 4 seconds in length each. The 60 recordings are broken down as 2 re-recordings each of 2 intensities each of 2 statements of 8 emotions. However, the "Neutral" emotion used only a single intensity. We split the dataset into 240 for calibration-validation, 840 samples for calibration, and 360 for testing using stratified

sampling over both emotion and gender. Given that each split was treated the same way they can be considered i.i.d. for conformal prediction. To process the audio samples, we used a wav2Vec2 model Baevski et al. (2020) available on Huggingface (Fadel, 2023). We resampled each file to 16kHz as the base model was pre-trained on that frequency. For audio with multiple channels, we changed it to mono by averaging channels. The model's feature extractor then turned the raw audio signals into feature vectors ready for the model to use, which were subsequently padded to ensure consistent input length for batch processing by the model. RAVDESS is released under a Creative Commons Attribution-NonCommercial-ShareAlike 4.0 International license.

## A.2 Conformal Prediction Methods and Hyperparameters

When generating conformal prediction sets for our human experiments, we aimed to diversify the settings by testing two different score functions. This helps demonstrate that our hypotheses and conclusions are not a result of using one particular score function. We chose two highly-performant methods for our tasks:

**RAPS** Regularized Adaptive Prediction Sets (Angelopoulos et al., 2021) builds off of APS (Romano et al., 2020b) by penalizing the score for sets with more elements than a threshold $k_{\text{reg}}$. In addition RAPS uses temperature scaling on the model's logits before the softmax layer with value $T$, and a weight $\lambda$ on the regularization term. First, RAPS defines $\rho_x(y) = \sum_{y'=1}^{M} f(x)_{y'} \mathbb{1}[f(x)_{y'} > f(x)_y]$ as the total probability mass of the labels which have higher softmax values than $y$ for input $x$, and $o_x(y) = |\{y' \in \mathcal{Y} \mid f(x)_{y'} \geq f(x)_y\}|$ as the ordinal ranking of $y$ among all labels, again based on softmax values. RAPS constructs prediction sets as

$$\mathcal{C}_{\hat{q}}(x) = \{y \mid \rho_x(y) + u \cdot f(x)_y + \lambda(o_x(y) - k_{\text{reg}})^+ \leq \hat{q}\}. \tag{12}$$

where $u \sim U[0, 1]$ is a uniform random variable. Here, the score function $s(x, y)$ has three terms that use the probability mass of classes more likely than $y$, the probability of $y$ with a random weighting, and a regularization term that penalizes adding more than $k_{\text{reg}}$ classes.

**SAPS** Sorted Adaptive Prediction Sets (Huang et al., 2024) is a more recent procedure that aims to produce compact prediction sets with better conditional coverage compared to other methods. SAPS has one primary hyperparameter, $\lambda$, which governs the weight given to the ranking information. As with RAPS, a temperature hyperparameter $T$ is optimized to scale logits before applying the softmax. SAPS discards all probability values except the maximum softmax probability, while preserving the ranking of the labels. The score function is defined as

$$s(x, y) = \begin{cases} u \cdot f(x)_y, & \text{if } o_x(y) = 1, \\ \max f(x) + \lambda(o_x(y) - 2 + u), & \text{otherwise,} \end{cases} \tag{13}$$

where $u \sim U[0, 1]$ is again a uniform random variable. This score function minimizes the effect of unreliable small softmax values while retaining enough ranking information to adjust the prediction set size based on instance difficulty.

**Hyperparameter Tuning** For prediction set construction, we used three splits of the data: calibration-validation $\mathcal{D}_{\text{calval}}$, calibration $\mathcal{D}_{\text{cal}}$, and test $\mathcal{D}_{\text{test}}$. $\mathcal{D}_{\text{calval}}$ was employed for hyperparameter tuning, $\mathcal{D}_{\text{cal}}$ was used to calculate conformal thresholds with the tuned hyperparameters, and finally $\mathcal{D}_{\text{test}}$ was used to generate the prediction sets used in our human experiments. Table 7 presents the final hyperparameters after automated tuning.

For hyperparameter optimization with CP methods we employed Bayesian optimization via the Optuna library (Akiba et al., 2019b), utilizing the TPESampler for efficient search over 50 iterations aiming to minimize average set size. Specifically, for each parameter setting, the conformal threshold $\hat{q}$ was determined using $\mathcal{D}_{\text{cal}}$, then prediction sets were generated on $\mathcal{D}_{\text{calval}}$ where their average size was computed. For Mondrian CP, within every iteration of tuning the same hyperparameter settings were used on each group.

Tuning for avg-$k$ is necessarily different because the main hyperparameter $k$ is the target average set size. Hence, instead of tuning for minimal average set size which would be trivial, we tuned $k$ to achieve the desired 90% coverage rate, matching the CP methods. $k$ was adjusted through a binary search by computing the threshold $q_k$ on $\mathcal{D}_{\text{cal}}$, then generating sets on $\mathcal{D}_{\text{calval}}$ and evaluating their empirical coverage. We refined the average set size $k$ through binary search up to five decimal points of precision to ensure that the coverage on $\mathcal{D}_{\text{calval}}$ closely matched the predefined target coverage.

Table 7: Hyperparameter Settings for Each Dataset After Tuning

| Dataset | Score Function | Marginal | | | Conditional | | |
|---|---|---|---|---|---|---|---|
| | | $T$ | $\lambda$ | $k_{\text{reg}}$ | $T$ | $\lambda$ | $k_{\text{reg}}$ |
| FACET | RAPS | 0.32 | 0.17 | 4 | 0.38 | 0.46 | 4 |
| BiosBias | SAPS | 0.53 | 0.21 | - | 0.53 | 0.26 | - |
| RAVDESS | RAPS | 0.26 | 10.49 | 3 | 0.20 | 0.43 | 3 |

## B  HUMAN SUBJECT EXPERIMENTS

In this section we give complete details on our experiments involving humans which extends the descriptions in Section 4 and Section 5.

**Participant Recruitment**    We designed our human subject experiments using PsychoPy (Peirce et al., 2019) and made them available on Pavlovia (Pavlovia, 2024). Participant recruitment was conducted through Prolific (Prolific, 2024). We required participants volunteering for the study to be fluent in English, the language we used throughout the experiments, and to use a desktop or laptop computer, but otherwise did not filter candidates on any criteria. In total, 600 unique participants were recruited, and each was randomly assigned to a single (task, treatment) pair, of which there were 12 in total for 50 participants each. Hence, each experiment involved a disjoint set of participants. Participants were paid according to the platform guidelines enforced by Prolific – participants were given a flat rate for completing their assigned experiment calculated based on the median completion time of their cohort. We also offered participants bonus pay in the amount of 0.01 GBP for every correct answer they gave on the 50 test trials, which incentivizes high quality answers. On average, participants received pay at a rate of 9.75 GBP/hour which totaled 1500 GBP in participant renumeration across all tests.

Across the experiments, 599 participants gave consent for their demographic data to be collected and shared in aggregate, while one participant withdrew their consent. As shown in Table 8, the study population is fairly balanced in terms of gender. Participants in the study represented various age groups and ethnicities, with more being in their 20s and identifying as White respectively.

Table 8: Demographics of Participants

| | Group | # Participants |
|---|---|---|
| **Age group** | < 20 | 31 |
| | 20-29 | 324 |
| | 30-39 | 144 |
| | 40-49 | 64 |
| | 50-59 | 26 |
| | $\geq 60$ | 10 |
| | Unknown | 1 |
| **Gender** | Female | 287 |
| | Male | 311 |
| | Unknown | 2 |
| **Ethnicity** | White | 346 |
| | Black | 179 |
| | Mixed | 34 |
| | Asian | 23 |
| | Other | 15 |
| | Unknown | 3 |

**Experiment Details**    Our experimental design follows that of Cresswell et al. (2024). An overview of the screens that were presented to participants is shown in Figure 7 using the BiosBias task with marginal treatment as an example. We will now walk through the progression of these screens, describing the test. For each experiment, participants were first shown a consent form detailing what data we were collecting, how we intended to use it, and how long we would retain it. Participants who did not consent could remove themselves from the study rather than proceeding. To limit

risks to participants, we did not collect any personally identifiable information like name, birth date, or address. As noted above, some demographic information had been provided by participants to the Prolific platform, and nearly all participants consented to us obtaining this information. We determined that there were no risks to participants taking the study that needed to be disclosed.

We proceeded to present participants with instructions about how to take the test and about the classification task they would be performing. We provided one labeled example datapoint from each class for participants to familiarize themselves with, and then conducted 20 practice trials that used the same format as the real test trials. For example, the main trial screen for BiosBias is shown in Figure 7, while Figure 3 shows FACET, and Figure 6 shows the RAVDESS task. The practice phase was for training participants to ensure they would achieve high accuracy on the real test trials, so we did not use data collected on the practice trials. Example and practice datapoints were sourced from $\mathcal{D}_{\text{test}}$, and were not reused for the real test trials. After the practice trials, 50 test trials were given, and the data collected here comprises what we used for analysis. For each task we randomly sampled sets of 50 datapoints from $\mathcal{D}_{\text{test}}$ without replacement using one of 10 fixed seeds. Each participant was shown one of these random sets. The use of only 10 seeds was intentional so that multiple participants across treatments would see the same datapoints. Since datapoints can have different inherent difficulty, using the same examples across treatments reduces variance from random data selection. Overlap between the examples shown to different participants also allows us to better model their inherent skill using our GEE analysis.

The main trial screens all show a participant one datapoint $x$, and all classes $\mathcal{Y}$ as options. Participants had unlimited time to select one class, after which the correct answer was shown following previous studies in machine learning (Stein et al., 2023). Providing answers after each trial has been shown to increase the quality of collected data (Mitra et al., 2015) and allows the participant to continually optimize their decision strategy. This is desirable, as we want participants to give the most accurate answers possible, even if their strategy evolves over the course of the test.

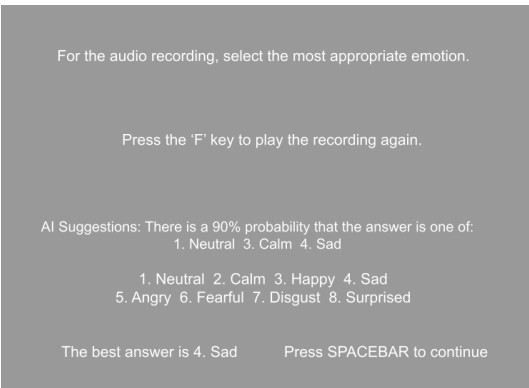

Figure 6: Main trial screen shown to participants for RAVDESS with marginal conformal set treatment. When the screen was first displayed, the audio recording was played, with the model suggestions appearing and ability to enter a response activated after the recording completed (roughly 4 seconds). Participants were able to replay the recording as many times as needed. The correct answer was shown after the participant responded.

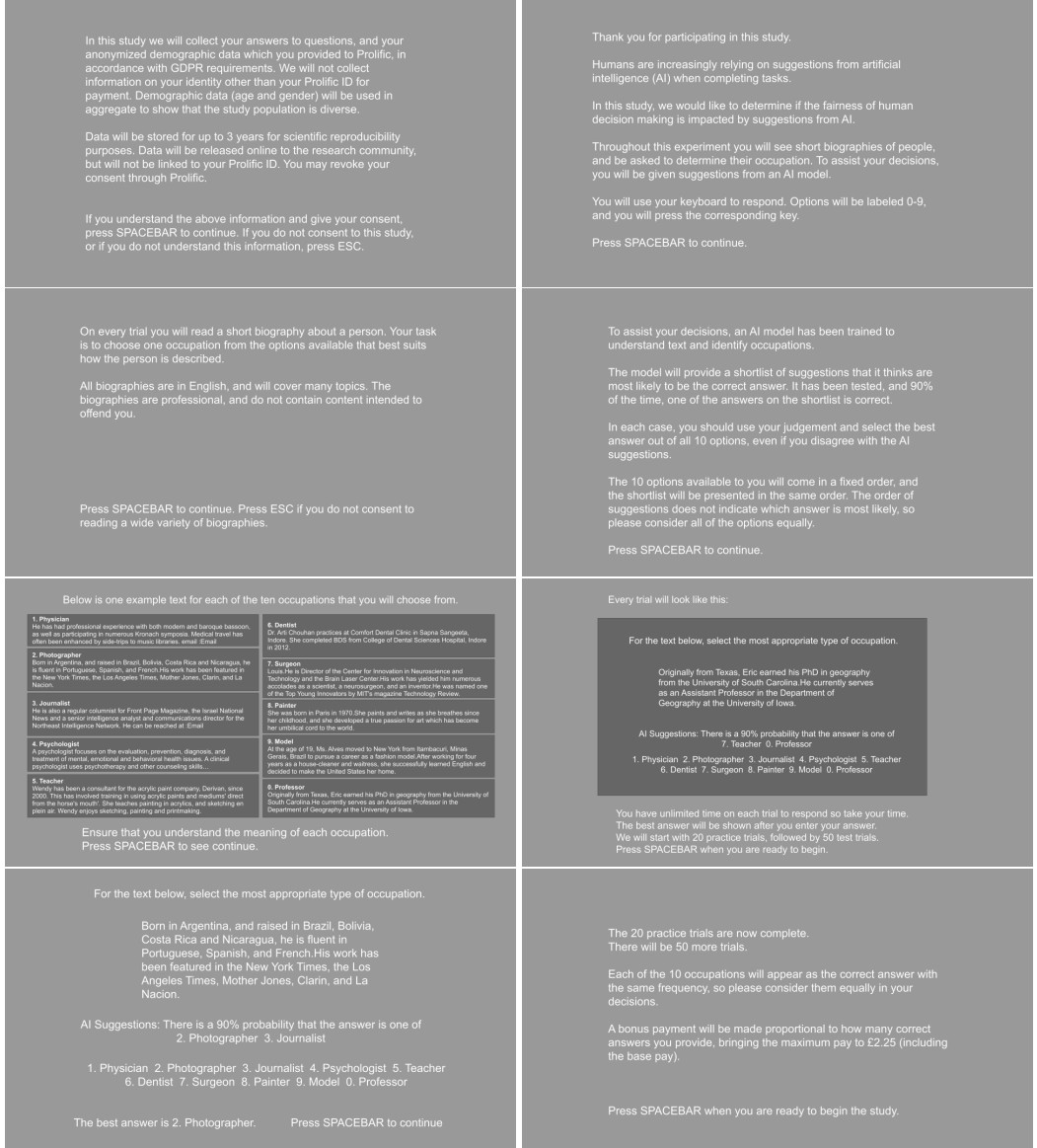

Figure 7: Screens displayed to participants during our experiment using marginal conformal sets on BiosBias, with a similar template for other experiments. The seventh panel (left to right, top to bottom) shows an example of how the practice and test trials appear, with the correct answer displayed after the participant entered their response. The fourth panel was not shown to Control group as they were not given suggestions from the model during the trials.

