# OpenReview forum: "Conformal Prediction Sets Can Cause Disparate Impact"
_ICLR.cc/2025/Conference — ICLR 2025 Spotlight_

### Official Review · Reviewer_EuTp · 2024-10-29

**Soundness:** 3
**Presentation:** 3
**Contribution:** 3
**Rating:** 8
**Confidence:** 3

**Summary:**

The paper presents a user study highlighting concerns regarding conformal prediction (CP) in practice. The authors show that $(i)$ CP can exacerbate fairness concerns when humans make decisions; $(ii)$ using equalized sets rather than equalized coverage (as previously proposed) leads to fairer outcomes in practice.

**Strengths:**

The main strengths of the paper are:

S1) the empirical evaluation is strong enough, involving different data types and settings;

S2) the proposed user study sheds light on some of the limitations of conformal prediction and provides an interesting counter-example to traditional approaches mitigating unfairness in conformal prediction;

S3) the work is overall well-written and easy to follow.

**Weaknesses:**

I do not have significant concerns regarding the work. I list here a few comments/suggestions regarding a few potential limitations worth discussing in the current version of the paper:

1) When considering selective prediction (see e.g., [1] for a survey), Bondi et al., (2022) [2] show that human performance is affected by how and which information is passed to them. Their experiments consider different settings and show that human performance increases when humans receive only the " deferral " information, and the ML model is good at solving the task. Similar considerations could also be taken into account in this setting.

2) As noted in De Toni et al. (2024), conformal prediction is mainly beneficial when the classification task regards several classes. When the task is binary, most methods fail to provide useful sets, as when both classes are returned, this does not help humans. I wonder whether this consideration might hinder the external validity of the paper's findings, as the current experiments concern only multiclass settings ($m\geq8$ for all three datasets).

3) I would be more cautious regarding lines 477-479. Ruggieri et al. (2023) [3] show that performing conditional corrections at the group level does not ensure an overall improvement in terms of fairness due to the Yule Effect. If my understanding is correct, generating more singletons per each protected group is equivalent to performing "conditional corrections"; hence, the Yule Effect might still occur. This means equalizing the number of singletons across groups might not always be the right solution.

[1] - Hendrickx, Kilian, Lorenzo Perini, Dries Van der Plas, Wannes Meert, and Jesse Davis. "Machine learning with a reject option: A survey." Machine Learning 113, no. 5 (2024): 3073-3110.
[2] - Bondi, Elizabeth, Raphael Koster, Hannah Sheahan, Martin Chadwick, Yoram Bachrach, Taylan Cemgil, Ulrich Paquet, and Krishnamurthy Dvijotham. "Role of human-AI interaction in selective prediction." In Proceedings of the AAAI Conference on Artificial Intelligence, vol. 36, no. 5, pp. 5286-5294. 2022.
[3] - Ruggieri, Salvatore, Jose M. Alvarez, Andrea Pugnana, Laura State and Franco Turini. "Can we trust fair-AI?." In Proceedings of the AAAI Conference on Artificial Intelligence, vol. 37, no. 13, pp. 15421-15430. 2023.

**Questions:**

I have a couple of questions for the authors:

q1) in the background section, I think the definition of conformal and non-conformal set predictors can coincide, depending on how $s()$, $f()$ and the relative quantiles are defined. Am I correct?

q2) regarding the previously highlighted shortcomings, can the authors elaborate on W1, W2 and W3?

---

> ### Author Response · Authors · 2024-11-16
> **Initial Response by Authors**
>
> Thank you for your effort in reviewing our paper. We appreciate that you found our experiments strong and diverse, that the paper was well-written, and that you were interested in how our experiments run counter to the prevailing wisdom. While you didn’t raise any significant concerns, we welcome your suggestions for improvement. We will make some comments on the the related papers you mentioned, and will incorporate them into our related works section.
>
> **1** Bondi et al., (2022) is another study involving humans but where a model can defer the final decision to a human. Hence, humans were only engaged on a small fraction of the test dataset, whereas we consider humans as the main decision makers for every example. You noted that Bondi et al. saw human accuracy increase when the deferral information was shared, but not when the model prediction was shared. This is perhaps unsurprising, since the only examples tested were ones where the model deferred, and these are exactly the cases where the model is most likely to be incorrect. Hence, we should not extrapolate such findings to the less constrained setting we used, where any example in the test set is given to a human, including cases where the model is highly confident and likely to help. The point you are trying to make remains valid though; the format of information provided to the user is likely to have a large effect on how helpful it is, but the overall setting can matter just as much. These are both aspects deserving of future research.
>
> **2** We noted De Toni et al. (2024) on page 1 as a prior work involving prediction sets and human decisions. De Toni et al. consider a constrained setting where experts (synthetic or human) must choose a class from the prediction set. This upper bounds the possible accuracy of experts to the coverage level of the sets. In our experiments, participants could choose any class and were free to use the prediction set information as they like. Both settings can be useful in different contexts. However, it does limit the transferability of conclusions between the works.
>
> We agree with your comments that conformal prediction is much less useful in binary settings since the range of possible sets lacks the diversity needed to express uncertainty. Essentially, uncertainty becomes a binary notion as well - the prediction set is either a singleton (confident), or one of $\\{0,1\\}$ and $\emptyset$ (not confident). In our reading of the literature, CP is rarely studied in the binary case because of its trivial nature. While we did not present a case study for $m=2$, our three examples with distinct values of $m>=8$ are more closely aligned with standard applications of CP, i.e. to multiclass problems.
>
> **3** This is a helpful comment to point out, and Ruggieri et al. (2023) give valuable guidance. We agree that group-conditional modifications, such as one would perform to generate more singleton sets, could be viewed as conditional corrections susceptible to the Yule Effect. Perhaps Mondrian CP as a whole could be viewed in this light, as it generates different conformal thresholds $\hat q$ per group which are meant to correct any groupwise miscoverage. We have not conclusively proven that equalizing the frequency of singletons leads to more fair outcomes. Across our three case studies, singleton frequency difference was correlated to disparate impact (Figure 4), but this is the extent of what we claim. Moreover, we did not propose methodologies that would equalize singleton frequency and then evaluate their fairness outcomes, but for example this could be done by setting the $k_\text{reg}=1$ in the RAPS algorithm (see Equation 12). Devising methods to equalize singleton frequency, and test them for fairness while avoiding adverse effects like Yule’s should be considered in future research.
>
> Importantly, none of the three papers above consider the fairness aspects of their human-AI decision systems, which is where our work breaks new ground.
>
> **Q1**
> > I think the definition of conformal and non-conformal set predictors can coincide... Am I correct?
>
> In short, yes. The conformal and non-conformal methods we discussed are not too different in spirit, both having a calibration step using a quantile, and adding classes to the prediction set based on that quantile. The main difference is that for conformal methods, the quantile is based on the desired coverage rate $1-\alpha$, whereas for Average-$k$ the quantile is based on the desired set size $k$. We could generalize the Average-$k$ method to use an arbitrary score function $s$ like CP does. Then, if the desired coverage and set size were carefully tuned such that the quantiles $\tfrac{\lceil{(n+1)(1-\alpha)}\rceil}{n}$ and $1-k/m$ were equal, the methods could coincide (see Sections 2.2 and 2.3 for notation). In practice, the score functions we used were quite different, with the Average-$k$ version (Algorithm 1) being much simpler than either RAPS or SAPS (Appendix A.2).

---

> > ### Comment · Reviewer_EuTp · 2024-11-17
> > **Response to author rebuttal**
> >
> > I thank the authors for elaborating on my comments and clarifying my doubts on Q1.
> > After reading other reviewers' comments and the authors' rebuttals, I am confident in keeping my current score, as I think this is a valuable contribution.

---

### Official Review · Reviewer_27ZU · 2024-10-31

**Soundness:** 3
**Presentation:** 3
**Contribution:** 3
**Rating:** 8
**Confidence:** 3

**Summary:**

This paper study conformal prediction setup, and want to ensure group fairness in outcome, as measured by disparate impact (final decision made). They study via human experiments, how two different fairness metrics on the prediction sets (fairness on the coverage or of the size) and show that i) the former worsens the actual fairness of the outcome and ii) the latter does improve it.

While I was not familiar with the conformal  prediction setup, I found the paper easy to read and it sparked my curiosity. From the research perspective, I believe that this paper is extremel valuabley to ensure that metrics are well grounded in actual outcomes.

**Strengths:**

As mentioned, I find that such connection between metrics on the set and end-behavior are needed. Paper is quite easy and claims are well supported by evidence.

**Weaknesses:**

Abstract could be a bit clearer, in particular for people less familiar with the subject.

**Questions:**

I am curious about the relationship between the Fairness in Conformal predictions and Fairness in Classifiers. Let's imagine a 3rd experiment, where the base classifier is fair and the conformal prediction setup is applied without any fairness constraint. What would the results look like>?

---

> ### Author Response · Authors · 2024-11-16
> **Initial Response by Authors**
>
> We are pleased to see that our work sparked your curiosity, and that you found it easy to read and extremely valuable research on fairness with well supported claims. We are also happy to address the few concerns you had.
>
> **W1**: Thank you for pointing out that our abstract could be made more accessible to a wider audience. We will add additional explanation so that it assumes less familiarity with conformal prediction and fairness.
>
> **Q1**
> > I am curious about the relationship between the Fairness in Conformal predictions and Fairness in Classifiers. Let's imagine a 3rd experiment, where the base classifier is fair and the conformal prediction setup is applied without any fairness constraint. What would the results look like?
>
> This is an interesting question which we didn’t directly address in the paper. To be clear, we are claiming that conformal prediction *can* cause disparate impact, not that it *always* causes disparate impact. We believe that the scenario you outline is one where disparate impact would not arise.
>
> For example, Figure 2 illustrates the mechanism of unfairness with CP. Notice that we are assuming the base classifier has some bias already (the model accuracy is higher on group $e$ than group $h$). Marginal CP tends to propagate this bias to human accuracy, and Conditional CP can amplify the bias further. However, if the base classifier is fair, and its behaviour does not have dependence on the group information, then there is no reason to believe that CP would generate unfairness. Model accuracy would be the same across groups, as would the coverage level and set size. Since there would be no difference in the properties of the prediction sets, human performance improvement should also be independent of group.

---

> > ### Comment · Reviewer_27ZU · 2024-11-21
> >
> > I thank the authors for their answer and the additional intuition they bring.

---

### Official Review · Reviewer_Ptux · 2024-10-31

**Soundness:** 3
**Presentation:** 3
**Contribution:** 3
**Rating:** 8
**Confidence:** 3

**Summary:**

The authors design a randomized trial to study the impact of utilizing conformal prediction (CP) sets in machine learning decision pipelines with a human in the loop. They find that utilizing a "fair" notion of CP (a notion which ensures coverage for each sensitive group) can actually exacerbate disparate impacts in the pipeline.

**Strengths:**

* The background on conformal prediction provided by the authors is well written, as are the justifications for the models picked at inference time.
* The authors pre-registered the hypothesis they tested
* I think the findings are generally novel and interesting. As figure 5 indicates, the fact that utilizing Mondrian CP can reduce accuracy for on certain groups has important implications for fairness practices when prediction sets are required.
* Generally, this feels like a paper that could spark some new work on fairness in CP, so I believe it will be of interest to the ICLR community.

**Weaknesses:**

* The authors primary focus is to study disparate impact as measured by improvement *gain* from a given CP treatment (as compared to a control). This is fine, but I would argue that if a CP treatment improves accuracy on each sub-group (even if at a different amount) then this isn't necessarily a bad thing, for example the minimax fairness would be improved. In this sense, I actually found figure 5 the most interesting observation, as it indicates how Mondrian CP can actually leave the worst-case groups, off worse. In general I would suggest more discussion on the trade-offs of the specific fairness definitions the authors use.

**Questions:**

* Could the authors provide further discussion on the implication of using conformal set size as a proxy for difficulty? I believe this is a reasonable proxy for difficulty but obviously it is also related to the treatment, does this have any effect on the analysis?
* A clarification question on equation 9. Does trial j refer to an entire task (e.g. text classification) or one sample from an entire task?

---

> ### Author Response · Authors · 2024-11-16
> **Initial Response by Authors (1/2)**
>
> Thank you for your time and effort spent reviewing our work. We were delighted to see that you found it “novel and interesting”, “well written”, and that it could “spark new work on fairness in conformal prediction”. Like you, we found the experimental results in Figure 5 surprising, since they run counter to the prevailing wisdom in the field around fairness for set-predictors. We will gladly address your concerns below.
>
> **W1**
> > … I would argue that if a CP treatment improves accuracy on each sub-group (even if at a different amount) then this isn't necessarily a bad thing, for example the minimax fairness would be improved.  In this sense, I actually found figure 5 the most interesting observation, as it indicates how Mondrian CP can actually leave the worst-case groups, off worse. In general I would suggest more discussion on the trade-offs of the specific fairness definitions the authors use.
>
> Fairness is a highly contextual notion; the correct fairness standard to apply depends entirely on the circumstances. We can certainly imagine some scenarios where a method would be beneficial if it improves accuracy on all groups, even if the amounts of improvement are not equal. For instance in a setting like medical treatment, it may be desirable to provide the most effective treatment on an individual basis. As long as no groups are receiving worse outcomes from a new type of treatment, we prefer to give every individual the best care possible. In this case, the disparate impact measure we applied would not be as relevant.
>
> Just as well, there are certainly scenarios where unequal improvements would lead to harms for some groups. Consider a realistic setting where resources are constrained, like mortgage lending. A bank has a fixed, finite amount of capital it can distribute as loans, and receiving a loan is a beneficial outcome for individuals. Since capital is constrained, giving a loan to one individual at the margin means not giving a loan to someone else. These people may belong to different groups, like high and low income. Now, the bank uses a model to generate prediction sets which are given to a human underwriter who then makes a decision whether to issue a loan. If the sets increase underwriter accuracy for all groups, but by different amounts, one group will benefit **at the expense of another**. If accuracy increases more for high-income earners than low-income, high-income earners may receive proportionally more loans at the margin. Even though accuracy increased for the low-income group, they received fewer loans and experienced worse outcomes. This is an example where fairness corresponds to minimizing disparate impact.
> As you pointed out, our results also showed one case where equalizing coverage of prediction sets actually reduced human accuracy compared to the control on some groups. This is a more serious problem, where it is abundantly clear that some groups are worse off when Mondrian CP is used.
>
> Given that fairness is such a nuanced topic, we do not expect a single definition to cover all cases. However, even in the title we emphasized that our perspective is about disparate impact. In Sections 1 and 2.4 especially, we were clear about the settings we consider, and the fairness definitions we focused on. We will improve this discussion by talking more about the nuances of fairness, and trade-offs of using a specific measure like disparate impact.

---

> > ### Author Response · Authors · 2024-11-16
> > **Initial Response by Authors (2/2)**
> >
> > **Q1**
> > > Could the authors provide further discussion on the implication of using conformal set size as a proxy for difficulty? I believe this is a reasonable proxy for difficulty but obviously it is also related to the treatment, does this have any effect on the analysis?
> >
> > As you say, we used marginal conformal set size as a proxy for the difficulty of each individual example in our statistical model (L299 and Equation 9). In the model it is important to include a covariate for difficulty, as each participant is shown a random subset of the test dataset, and so we need to account for the variability of difficulty across subsets. Marginal set size is a measure of **inherent** difficulty, independent of the treatment that is applied to participants. The same proxy for difficulty was used in the model regardless of what treatment was applied to the participant. We accounted for the effect of treatment separately via the $\mathrm{treat}_{i}$ covariate, and randomization of treatment assignment ensures that estimated treatment effects are not confounded by difficulty. Moreover, individuals may find given examples more or less difficult than the population due to personal characteristics. For example, on the image classification task a person who is color blind would have a different experience than a person with typical color vision. However, the statistical model (GEE) already accounts for individual effects. The covariate for difficulty is therefore meant to capture the inherent difficulty of each datapoint.
> >
> > There is evidence that supports the reasonableness of this proxy and suggests that examples which humans find difficult on their own are the same examples which models have low accuracy on, at least for the types of tasks we are considering. Figure 5 shows a direct comparison between human accuracy and model Top-1 accuracy by group for FACET. The trends of which groups have higher or lower accuracy are the same. More directly though, past work has this effect in more detail; our design choice was informed by Figure 8 of Cresswell et al. 2024, where it was shown that Control group accuracy decreases as marginal set size increases, which we find to be strong evidence that the proxy is reasonable.
> >
> >
> > **Q2**
> > > A clarification question on equation 9. Does trial j refer to an entire task (e.g. text classification) or one sample from an entire task?
> >
> > In Equation 9, trial $j$ refers to an individual datapoint shown to a human. Each participant $i$ was tested on 50 different trials $j$. We use “task” to refer to the combination of dataset and treatment, such that each unique participant in the study was only tested on a single task.

---

> ### Comment · Reviewer_Ptux · 2024-11-21
>
> Thank you for the interesting answers to my questions, I maintain my score.

---

### Official Review · Reviewer_5hcH · 2024-11-01

**Soundness:** 2
**Presentation:** 3
**Contribution:** 2
**Rating:** 5
**Confidence:** 3

**Summary:**

This work is motivated by the well-known problem of group-conditional coverage in conformal prediction: when an underlying predictive model is more inaccurate on some subgroups than others, achieving equalized coverage over subgroups requires providing prediction sets that vary in size across subgroups. The work conducts a human study in which human decision-makers are presented with prediction sets as a decision aid; the study suggests that their accuracy improves at different rates across subgroups.

**Strengths:**

The authors rightly identify "equal coverage" as a limited notion of fairness for prediction sets, in the sense that the end-goal of providing prediction sets is ultimately to improve a downstream task -- i.e., equalized coverage in itself ought to be desirable only to the extent that it improves the utility of predictions in general. Evaluation via human study is also an important perspective; I see this work as playing a similar role as the work that conducted human evaluations of explainability methods. I also appreciate that authors mention (non conformal) methods for providing prediction sets.

**Weaknesses:**

* The main weakness of this work to me is that its hypotheses are a priori unsurprising. If some subgroup A requires larger set sizes for the same coverage level, it is tautologically true that there is higher uncertainty for subgroup A and that any class in the prediction set is less likely to be the ground-truth label. Even if a person relied 100% on the prediction set and simply picked uniformly at random among the predictions, one would expect performance to be worse on groups with bigger set sizes.
* Therefore, I find the results to be less 'disquieting' than the work seems to suggest. Over the baseline of no assistance, it seems that all methods approximately improve accuracy, and I don't really see why it's a bad thing that some groups benefit more; the paper does not make a case _against_ using conditional CP methods for bias reasons. (That is, why artificially reduce accuracy, i.e. worsen outcomes, for the easier subgroups? It's not clear to me how that would be normatively defensible, e.g. in a high stakes medical task, and I would push back on the language of "increases unfairness".) It would certainly be a bad thing if conditional CP somehow made human prediction worse than marginal CP; a different question that could/would have been interesting to ask is not whether max disparity increases, but rather whether accuracy actually improves from seeing conditional prediction sets or not. That is, something like H0: for all subgroups, accuracy using marginal and accuracy using conditional prediction sets is the same; H1: for at least some subgroups accuracy increases or decreases.
* It is also surprising to me that the baseline is no assistance rather than providing a point (top-1) prediction; the latter seems like the more natural comparison for a prediction set and could potentially illustrate more significant per-subgroup disparities, especially with more variation in top-1 accuracy.
* Finally, minor comment on experimental design - I should preface this comment by saying I'm not an expert in this kind of study design (though I am familiar with other approaches to randomized experiments and hypothesis testing). However, I am not confident that the numerical results (i.e. those summarized in Table 2) lend themselves to clear statistical interpretations - e.g., given the computed point-estimate ORs, how are we meant to interpret (max) ROR, and where does that uncertainty propagate? How are significance thresholds computed? This is especially concerning given that for FACET, the only dataset where set sizes differ by more than 1, all the ORs themselves essentially all hover around 1/ just barely at significance. That said, I think the primary value of this work is the high level characterization of how humans use and interpret prediction sets (as opposed to applications of RCTs in econ/medicine/etc, where precise interpretations and therefore rigorous experimental design are more consequential).

======

Minor presentation notes:

* Exchangeability - should probably be formally defined. Saying that it is "realistic in practice" (086) is a pretty strong statement, in my opinion. I personally consider the exchangeability/iid distinction to be more formal than practical - I've very rarely seen practical applications that are exchangeable but not iid. This is not central to the claims of the paper and I'm not necessarily counting it as a weakness; I just found that statement to be distracting.
* Eq 8 - assuming `Cov` means `Coverage` but would probably be nice to define it formally first
* L241 - minor typo in this sentence I think.

**Questions:**

* Fig 1: what is the x-axis?
* Table 1: is the "coverage" column for the conditional method computed on average?
* Table 2: significance levels are for the null of OR = 1?

---

> ### Author Response · Authors · 2024-11-16
> **Initial Response by Authors (1/3)**
>
> We would first like to thank the reviewer for their time spent reviewing our paper and their helpful feedback. We appreciate that you found our work provides an “important perspective” on conformal prediction via human study, and that you agree with our conclusion that equalized coverage has limitations. We understood your main concerns to be around the significance of our hypotheses, the standard of fairness we used, and some aspects of the experimental design. We will address all these aspects below.
>
> **W1**:
> > The main weakness of this work to me is that its hypotheses are a priori unsurprising. If some subgroup A requires larger set sizes for the same coverage level, it is tautologically true that there is higher uncertainty for subgroup A and that any class in the prediction set is less likely to be the ground-truth label. Even if a person relied 100% on the prediction set and simply picked uniformly at random among the predictions, one would expect performance to be worse on groups with bigger set sizes.
>
> While we do not disagree with this hypothetical, the example is overly reductive and misses a crucial point of our work. In reality, people using prediction sets as a decision aid **do not** pick an answer from the set randomly - the hypothetical argument could be a distraction. For example, we have found in practice that decision makers sometimes select a label that was not in the prediction set at all (Section 6.2; Adoption Rate). The way that humans interact with and use prediction sets is complex and we do not yet have a full understanding of it. The works we reviewed in Section 1 have started to investigate these interactions. Given that, without controlled experimentation we cannot say how human performance will look on groups with larger or smaller set sizes.
>
> One of the main points we investigated and provided evidence for was the very notion that human performance will be higher when sets are smaller. While this would obviously be true for the random selection in your example, it might not be true for human decision makers in reality. One possibility is that human performance is more influenced by the coverage of sets than their size, i.e. it is more helpful if the prediction set often contains the correct answer, regardless of whether several incorrect labels are also in the set. If this were the case, then equalizing coverage would lead to more fair outcomes. It is not obvious *a priori* which factors of a set-predictor will have the biggest influence on users. In order to make claims one way or the other, we need scientific experimentation. This is what we investigated in Section 6.2 and Figure 4. We analyzed four factors which could potentially influence the fairness of outcomes and found that while set size is strongly correlated with performance, coverage is not. While you may have expected this outcome based on your correct intuition, you also have the benefit of hindsight. Other explanations were plausible before our experiments provided the hard data that now supports your intuition.
>
> Consider a practitioner who is new to the field of conformal prediction. The prevailing wisdom in the field since 2020 has been that equalizing coverage is a fair way to produce prediction sets (i.e. since the seminal work by Romano, Foygel Barber, Sabatti, Candès 2020). This has been repeated and reinforced in papers since, including recently in 2024 (see discussion starting L183). There is a real possibility that well-meaning practitioners see this research and deploy it in practice without the same nuanced understanding you have developed, but this would likely lead to increased unfairness as in our case studies. Our work serves as a public warning to consider the downstream fairness impacts of common design choices in conformal prediction.

---

> > ### Author Response · Authors · 2024-11-16
> > **Initial Response by Authors (2/3)**
> >
> > **W2**
> > > … it seems that all methods approximately improve accuracy ... I don't really see why it's a bad thing that some groups benefit more... It would certainly be a bad thing if conditional CP somehow made human prediction worse than marginal CP
> >
> > It is not the case that all set-predictor methods improved accuracy for all groups. We demonstrated cases where conditional CP led to worse performance than marginal CP, and **worse performance than the control with no model assistance**. Figure 5 (Right) shows how the human performance changed compared to the control for the four groups within FACET over the three treatments. Notably, the Conditional treatment led to worse outcomes for the Older and Unknown groups compared to the Control.
> >
> > We would also push back on the notion that as long as a method $M$ improves accuracy on all groups by some (possibly different) amounts, it is a good thing. Fairness is a highly contextual notion; the correct fairness standard to apply depends entirely on the circumstances. In your example of a high stakes medical task, it may be desirable to provide the best possible care on an individual basis. In such a case, method $M$ should be applied, because regardless of which group the individual belongs to, $M$ will improve their outcomes. But this example makes the assumption that resources are effectively infinite -- that providing better care to an individual (via $M$) does not reduce the amount of care that any other person receives.
> >
> > Consider instead a realistic setting where resources are constrained, like mortgage lending. A bank has a fixed, finite amount of capital it can distribute as loans, and receiving a loan is a beneficial outcome for individuals. Since capital is constrained, giving a loan to one individual at the margin means not giving a loan to someone else. These people may belong to different groups, like high and low income. Now, the bank uses a model to generate prediction sets which are given to a human underwriter who then makes a decision whether to issue a loan. As in your example, if the sets increase underwriter accuracy for all groups, but by different amounts, one group will benefit **at the expense of another**. If accuracy increases more for high-income earners than low-income, high-income earners may receive proportionally more loans at the margin. Even though accuracy increased for the low-income group, they received fewer loans and experienced worse outcomes. This is an example of disparate impact, the main fairness quantity we study.
> >
> > > a different question that would have been interesting to ask is ... whether accuracy actually improves from seeing conditional prediction sets or not...
> >
> > We do not know of prior work that specifically focuses on whether conditional sets improve accuracy of humans compared to marginal sets. None of the studies we mentioned on L050 used conditional sets, so in that respect our work is the first. Although our presentation was centered around fairness using disparate impact as the measure, we can still find answers to your question in Table 2 and Figure 5. Table 2 shows odds ratios $\mathbf{OR}_{t,a}$, a measure of how much human performance improved on group $a$ under treatment $t$, compared to the Control. Larger values indicate more improvement. By eye, there are cases where the conditional treatment actually harmed performance on a group whereas the marginal treatment helped it. If you prefer plain accuracy values rather than odds ratios, Figure 5 leads to similar conclusions. In one case study (FACET), marginal improved accuracy on all groups, whereas conditional harmed accuracy on some.
> >
> > **W3**
> > > It is also surprising to me that the baseline is no assistance rather than providing a point (top-1) prediction...
> >
> > Providing top-1 predictions could be a sensible baseline, but our Control treatment with no model assistance has a distinct advantage. We take the notion of a set literally, and provide prediction sets without any ordering information (e.g. providing the ranking by model logits, or providing softmax probabilities). If top-1 predictions were given along with the prediction set, it would break the symmetry across options in the set. This could be an additional confounding variable which would make it more challenging to extract the effect of set size vs. coverage as we focused on. Additionally, we perform analysis in Figure 5 (Left) which considers the relative difficulty between groups as experienced by the model on its own (top-1 accuracy), and humans on their own (Control treatment). This helps to demonstrate how unfairness can arise. The groups which are hard for humans are also hard for the model, which means providing model information is least helpful for the groups that need the most help. We could not make such connections if the baseline used top-1 predictions.
> >
> > The baseline of no model assistance has been used in prior work, e.g. Zhang et al. 2024 or Cresswell et al. 2024 from our References.

---

> > > ### Author Response · Authors · 2024-11-16
> > > **Initial Response by Authors (3/3)**
> > >
> > > **W4**
> > >
> > > > …  given the computed point-estimate ORs, how are we meant to interpret (max) ROR, and where does that uncertainty propagate? How are significance thresholds computed? This is especially concerning given that for FACET, the only dataset where set sizes differ by more than 1, all the ORs themselves essentially all hover around 1/ just barely at significance. …
> > >
> > > We discussed the interpretations of ORs and (max) ROR in L317 and L325. An $\mathbf{OR}_{t,a}$ greater than 1 means group $a$ had higher odds of correct answers under treatment $t$ compared to Control, and less than 1 indicates worse odds. Comparing OR values for two groups tells us if one group benefited more than another under the same treatment.
> > >
> > > $\mathbf{ROR}_{t,a, b}>1$ means group $a$ experienced a larger increase in odds of correctness than group $b$, which is an unfair outcome under the definition of disparate impact. Significance values for ORs are output by the statistical model we used, described in Equation 9. The GEE implementation in the statsmodels package mentioned in L323 directly outputs $p$-values for the ORs. RORs are computed simply as the ratio of ORs. We intended maxROR as a descriptive statistic to give the reader a summary comparison of estimated ORs through a single metric, and as such we did not attempt to propagate the significance values to RORs. We observed disparate impact across three datasets (tasks), where some ORs are significant, while others are not. However, the consistent direction of the estimates aligned with our initial hypotheses. In scientific experiments, especially for baseline studies with limited sample sizes, we prioritize the direction of the effects over $p$-values, as these provide practical insight into potential biases. Rather than focusing on statistical significance with arbitrary thresholds, we emphasize the cumulative evidence of consistent directional estimates across different datasets and the real-world implications of disparate impact. However, in the spirit of transparency, we reported significance at the 5% and 10% levels.
> > >
> > > FACET has a few considerations that make it different from BiosBias and RAVDESS. For one thing, FACET has more than two groups and the populations are not balanced (Table 4). Due to the imbalance, the underrepresented groups were shown to humans less often, resulting in less data collected about them. This impacted the power of statistical analysis we could perform, and resulted in less statistically significant results, which we raised as a limitation in L528. Although the ORs are closer to 1 than for BiosBias and RAVDESS, a value of 0.91 for example still indicates that humans had 9% lower odds to give the correct answer compared to the control, which on its own could be concerning. When noting that another group improved their odds by 37% under the same treatment, the fairness concern should be clear. Additionally, the fact that FACET had the largest set size difference between groups (Table 1) may simply be because it had the highest number of classes (Table 3, $m=20$ compared to 10 and 8). We do not attribute any special significance to set sizes differing by more than 1 on average.
> > >
> > > **Minor presentation notes**
> > > > Exchangeability - should probably be formally defined. Saying that it is "realistic in practice" (086) is a pretty strong statement, in my opinion. I personally consider the exchangeability/iid distinction to be more formal than practical - I've very rarely seen practical applications that are exchangeable but not iid…
> > >
> > > We will happily include a formal definition of exchangeability, as it is a much less commonly discussed assumption than i.i.d. When we said it is “realistic in practice” on L086, we were referring to how it is a weaker assumption than i.i.d., which is itself quite commonly assumed. If one is happy to assume their data is i.i.d., then exchangeability is a given. We agree with you that the distinction is “more formal than practical”, and that it is rare to find cases which are exchangeable but not i.i.d. But from our perspective, exchangeability is realistic since it follows automatically from the stronger (but also realistic) assumption of i.i.d.
> > >
> > > >Eq 8 - assuming Cov means Coverage but would probably be nice to define it formally first
> > >
> > > Correct, we will add this definition to Equation 1.

---

> ### Comment · Reviewer_5hcH · 2024-11-20
> **Rebuttal response**
>
> Thanks for the engagement with the review here; my qualms with the work remain after rebuttal, and I will maintain my score. Of course it looks like this will largely be a symbolic action since this paper will probably get in the conference based on the other scores.
>
> While I understand there’s no way to re-do the research question and experimental setup from scratch, I hope that this feedback will be a useful signal for how some readers might receive the work given the current exposition. My strongest piece of actionable feedback at this point is to really walk back some of the more dramatic claims about increasing unfairness, and instead being more explicit about the nuances of the actual results. I appreciate that the authors repeatedly mention that ‘fairness is complex’ throughout the rebuttal, and some of the examples given in the rebuttal are thoughtful. But in my reading, the way that the current submission is actually written reflects a pretty reductive “numbers not equal == always really bad” perspective that actually obscures the real takeaway.
>
> I think that this work has potential to be quite impactful in shaping the high-level conversation (and, to the rebuttal’s point, a nonexpert’s perception of what is true) about conformal x fairness. Given that, I would really encourage the authors to do some revisions on the writing level, especially in the intro/abstract, to avoid misleading readers.
>
> (Also I still don’t understand Figure 1…)

---

> ### Author Response · Authors · 2024-11-24
> **Continued Discussion (1/2)**
>
> Thank you for your quick feedback and your kind words that the work could be impactful for the discussion of fairness in conformal prediction! It seems like we have come to an agreement on some points, but you still have qualms about the portrayal of some results. For instance we agree that:
> Fairness is a complex notion, and different situations require different fairness standards and measurements.
> Our main result that conditional conformal prediction actually can cause a certain type of unfairness (disparate impact) is surprising based on the prevailing wisdom in the literature.
>
> However, you feel that some of our claims lacked nuance, while noting your interpretation that our discussion seemed reductive and summarized it as “numbers not equal == always really bad”. While we do not agree with this description of our work, we appreciate your opinion and acknowledge that you are trying to help improve the work. Regardless of the eventual outcome of accept/reject, we hope to engage further to understand your perspective and modify the paper in a way that more clearly communicates the findings.
>
> You first recommended that we “walk back some of the more dramatic claims about increasing unfairness” in the abstract and introduction. Based on your initial review, we interpret your position as being about how we applied only disparate impact as the standard of fairness to our results, as compared to other any other form of unfairness such as your setting where increasing accuracy on different groups by different amounts should be viewed as beneficial (as long as no group sees accuracy decrease). Your concern is that it is overly reductive to only consider one type of unfairness, while boiling down judgements on fairness to numerical comparisons. However, a singular focus on disparate impact is very common in the fairness literature (e.g. references [A-E]) because it is widely applied in legal and regulatory systems that judge fairness in the real world. For example, in the United States the primary regulatory body for the banking industry (the OCC) uses disparate impact as its main criteria for determining if a bank’s policies and practices are fair [F]. Our work is perhaps most similar in approach to [B], which takes an interesting method from machine learning (differentially private SGD, which aims to improve privacy), and shows via case studies that it can cause disparate impact. [B] has had enormous impact on the field of privacy in machine learning by raising these concerns, but notably only evaluated fairness in terms of disparate impact, and did so numerically with a definition akin to our Equation 7. Hence, following all these works [A-E], our analysis was framed around evaluating disparate impact, and under this definition it simply is the case that having different values of $\mathbf{OR}_{t,a}$ for different groups $a$ is undesirable.
>
> Still, we are trying to take your feedback to heart, so we will revise the abstract and introduction to refer more specifically to “disparate impact” than “fairness” which is a more general concept. For instance in the abstract we will change:
> - “... prediction sets can increase the unfairness of their decisions.” -> “... prediction sets can lead to disparate impact.”
> - “... Equalized Coverage actually increases unfairness …” -> “... Equalized Coverage actually increases disparate impact …”
> - “... which empirically leads to more fair outcomes.” -> “... which empirically leads to lower disparate impact.”
>
> [A] Chouldechova. Fair prediction with disparate impact: A study of bias in recidivism prediction instruments. Big Data 5 2, 2017.\
> [B] Bagdasaryan et al. Differential Privacy Has Disparate Impact on Model Accuracy. NeurIPS 2019.\
> [C] Feldman et al. Certifying and removing disparate impact. SIGKDD 2015\
> [D] Zafar et al. Fairness Constraints: Mechanisms for Fair Classification. AISTATS 2017\
> [E] Zemel et al. Learning Fair Representations. ICML 2013\
> [F] Office of the Comptroller of the Currency. [Fair Lending (link)](https://www.occ.treas.gov/topics/consumers-and-communities/consumer-protection/fair-lending/index-fair-lending.html)

---

> > ### Author Response · Authors · 2024-11-24
> > **Continued Discussion (2/2)**
> >
> > You also recommended that we be “more explicit about the nuances of the actual results”. Because it is important to provide nuance when it comes to fairness, we discussed in Section 2.4 the distinction between “procedural and substantive fairness” (L157), and how “Equalized Coverage aligns with the concept of procedural fairness” (L189). We referenced how “substantive fairness focuses on outcomes” (L162), and described to the reader exactly how we were applying the notion of substantive fairness through the “change in per-group accuracy when prediction sets are supplied compared to unassisted decisions” (L168), i.e. “disparate impact” as in Eq. 7. As we have both acknowledged, there are other ways to approach substantive fairness, and we mentioned two other fairness standards that fall into this category, “Equalized Odds and Equalized Opportunity” (L164). Still, as with past work on fairness in ML [A-E], we focused on one concrete metric in our numerical evaluations, disparate impact. You have acknowledged that at this point it is not feasible for us to revise the paper to use a different metric.
> >
> > In Section 6.1 we discussed our main results in support of Hypothesis 1. For instance, we acknowledged that in “most cases prediction sets were useful to participants” (L419) as you have argued is important, but also pointed out that “the beneficial effect is not experienced equally across groups” (L421). Our main conclusions were framed around the specific measure of disparate impact, rather than using broader terms like “fairness” (L425, 428). We also acknowledged that “disparate impact need not always occur” (L429) with the RAVDESS example, which hearkens back to our paper’s title, emphasis on the word *can*, which makes it clear to the reader that we are not making unjustified blanket claims about CP *always* causing unfair outcomes. As for Hypothesis 2, our three case studies were entirely consistent, and our discussion starting L447 is clearly phrased in terms of disparate impact, rather than any more broad language about unfairness.
> >
> > Having reviewed the presentation of our main results at your suggestion, we have not found specific instances where the discussion could mislead the reader about what type of fairness is being considered, since we consistently refer to disparate impact. If you can point to specific lines in the paper where you interpret the claims as being misleading, we will happily clarify them.
> >
> > Finally, at your prompting we have revisited Figure 1 and understood that its clarity can be improved. It shows the main results from our three experiments in a way that is more accessible than the rigorous statistical analysis we described in Section 4.2 and Table 2. We believe the GEE model is the best way to interpret our results as it accounts for factors such as individual ability and inherent difficulty of data subsamples, but we also wanted to communicate a much more transparent analysis of the data based directly on human accuracy values. The $y$-axis shows the disparate impact measure which is formally defined in Eq. 7. Disparate impact is a property of the treatment $t$ - since there are three treatments, we have three disparate impact values per experiment. The $x$-axis is a categorical variable indicating which treatment was applied. From the data we might conclude that:
> > a) for two out of three datasets the disparate impact values for the Marginal CP sets were non-zero (and similar in magnitude to the Avg-$k$ treatment), which indicates that some groups benefited more than others;
> > b) for all three datasets the disparate impact was much higher for Conditional CP sets.
> > We prefer to use the GEE model to draw conclusions in our work, but in the end the conclusions would be the same.
> >
> > To improve the clarity of the figure we will:
> > - Move the dataset labels off the $x$-axis and instead put them above the respective subplots, with an indication for the reader that FACET, BiosBias, and RAVDESS are datasets as opposed to anything else.
> > - Label the $x$-axis for each subplot as “Treatment”, with the color-coded values still present in the Legend.
> > - Reference Equation 7 in the figure caption as the mathematical definition of accuracy disparate impact.
> >
> > We value your feedback, and have made a genuine attempt to understand your viewpoint. We hope that the changes we have suggested here to revise the writing will avoid any possibility of misleading readers as to what is being claimed about the fairness or unfairness of applying CP sets. Given that you do see the potential impact of our work, and the value of raising potential concerns to the community, we ask that you consider raising your score.

---

> > > ### Author Response · Authors · 2024-12-03
> > > **Reminder to Reviewer 5hcH**
> > >
> > > As the discussion period is ending promptly, we kindly ask if the reviewer has had time to consider our response? We would be happy to engage in discussion in the limited time remaining.

---

### Meta-Review · Area_Chair_Hgmp · 2024-12-09

**Metareview:**

The paper studies the fairness implications of conformal prediction (CP) sets when used in AI-assisted decision-making pipelines. It demonstrates that equalized coverage, a standard fairness metric in CP, can unintentionally increase disparate impact. The authors propose an alternative approach, equalized set size, which empirically reduces disparate impact in their experiments. Through human subject studies involving real-world datasets, the paper highlights how conformal prediction fairness metrics can influence downstream decision-making in unexpected ways.

The reviewers agree that the paper is timely and relevant and also views positively on the empirical evaluation through human subject experiments. The main strengths of the paper include novel insights into fairness trade-offs in CP and a well-designed experimental setup. However, the reviewers also expressed concerns about potential over-interpretation of findings, limited exploration of alternative fairness definitions, and framing of key claims. Overall, I find the paper’s focus on human decision-making, CP, and fairness to be an interesting and potentially impactful angle. Given the largely positive feedback from most reviewers, I lean towards recommending acceptance. However, I would also strongly encourage the authors to address the reviewers’ concerns, particularly those raised by 5hcH, in their final version if the paper is accepted.

**Additional Comments On Reviewer Discussion:**

The feedback is positive from most reviewers, though reviewer 5hcH has raised concerns regarding the framing of the paper. Given that the paper’s focus on human decision-making, CP, and fairness is interesting and potentially impactful, I lean towards recommending acceptance but encourage the authors to incorporate the reviewer's comments in the revision.

---

### Decision · Program_Chairs · 2025-01-22

Accept (Spotlight)